# Logarithmic-Regret Quantum Learning Algorithms for Zero-Sum Games

**Minbo Gao**
State Key Laboratory of Computer Science,
Institute of Software, Chinese Academy of Sciences,
Beijing, China
and University of Chinese Academy of Sciences,
Beijing, China
gmb17@tsinghua.org.cn

**Zhengfeng Ji**
Department of Computer Science
and Technology,
Tsinghua University,
Beijing, China
jizhengfeng@tsinghua.edu.cn

**Tongyang Li**
Center on Frontiers of Computing Studies,
and School of Computer Science,
Peking University,
Beijing, China
tongyangli@pku.edu.cn

**Qisheng Wang**
Graduate School of Mathematics,
Nagoya University,
Nagoya, Japan
QishengWang1994@gmail.com

## Abstract

We propose the first online quantum algorithm for solving zero-sum games with $\widetilde{O}(1)$ regret under the game setting.[1] Moreover, our quantum algorithm computes an $\varepsilon$-approximate Nash equilibrium of an $m \times n$ matrix zero-sum game in quantum time $\widetilde{O}(\sqrt{m+n}/\varepsilon^{2.5})$. Our algorithm uses standard quantum inputs and generates classical outputs with succinct descriptions, facilitating end-to-end applications. Technically, our online quantum algorithm "quantizes" classical algorithms based on the *optimistic* multiplicative weight update method. At the heart of our algorithm is a fast quantum multi-sampling procedure for the Gibbs sampling problem, which may be of independent interest.

## 1 Introduction

Nash equilibrium is one of the most important concepts in game theory. It characterizes and predicts rational agents' behaviors in non-cooperative games, finding a vast host of applications ranging from analyzing wars [45] and designing auctions [35], to optimizing networks [43].

It was shown in Daskalakis et al. [16], Chen and Deng [12] that finding a Nash equilibrium is PPAD-hard for general games. Nevertheless, computing the Nash equilibrium for specific types of games, such as zero-sum games, is particularly interesting. A zero-sum game requires that the utility of one player is the opposite of the other's, a condition that often appears in, for example, chess games. Von Neumann's minimax theorem [51] promises that every finite two-player zero-sum game has optimal mixed strategies.

---

[1]Throughout this paper, $\widetilde{O}(\cdot)$ suppresses polylogarithmic factors such as $\log(n)$ and $\log(1/\varepsilon)$, and $O^*(\cdot)$ hides quasi-polylogarithmic factors such as $n^{o(1)}$ and $(1/\varepsilon)^{o(1)}$.

37th Conference on Neural Information Processing Systems (NeurIPS 2023).

**Zero-Sum Games.** For a two-player zero-sum game represented by an $m \times n$ matrix $\mathbf{A}$, the Nash equilibrium is the solution pair $(x, y)$ to the following min-max problem:

$$\min_{x \in \Delta_m} \max_{y \in \Delta_n} x^\mathsf{T} \mathbf{A} y,$$

where $x \in \Delta_m$ and $y \in \Delta_n$ are $m$- and $n$-dimensional probability distributions, respectively. Usually, we are satisfied with an approximate Nash equilibrium rather than an exact one. An $\varepsilon$-approximate Nash equilibrium is a solution pair $(x, y)$ such that:

$$\max_{y' \in \Delta_n} x^\mathsf{T} \mathbf{A} y' - \min_{x' \in \Delta_m} x'^\mathsf{T} \mathbf{A} y \leq \varepsilon.$$

**Online Learning.** Since the matrix $\mathbf{A}$ of the zero-sum game usually has a large dimension in practice, it is common that we trade accuracy for space and time efficiency. Thus, online learning becomes increasingly significant in these scenarios. Online learning studies the situation when data is only available in sequential order and aims at making good decisions in this setup. In evaluating online learning algorithms, regret is an important criterion that measures how good an algorithm is compared with the optimal static loss (see more details in Section 2.3).

The idea of the online learning algorithms for zero-sum games stems from repeated play in game theory, e.g., fictitious play [11]. Specifically, we simulate the actions of two players for multiple rounds. In each round, players make decisions using a no-regret learning algorithm, considering the opponent's previous actions. For example, a famous algorithm of this type was proposed in Grigoriadis and Khachiyan [22] inspired by the exponential Hedge algorithm. The algorithm has regret $\widetilde{O}(\sqrt{T})$ and $T$ rounds, establishing the convergence rate of $\widetilde{O}(1/\sqrt{T})$.

It takes about two decades before the next improvement in Daskalakis et al. [17] to happen, where the authors proposed a strongly-uncoupled algorithm, achieving $\widetilde{O}(1)$ total regret if both players use the algorithm. They used the technique of minimizing non-smooth functions using smoothed approximations proposed in Nesterov [37], and this technique was later developed in Nesterov [38], Nemirovski [36] for broader classes of problems. Later, it was found in Syrgkanis et al. [46] that the optimistic multiplicative weight algorithm also leads to $\widetilde{O}(1)$ total regret with regret bounded by variation in utilities; this algorithm was recently extended to correlated equilibria in multi-player general-sum games in Anagnostides et al. [3]. It was proved in Hsieh et al. [30] that optimistic mirror descent with a time-varying learning rate can also achieve $\widetilde{O}(1)$ total regret for multi-players. Our quantum algorithm follows the overall idea of the optimistic multiplicative weight update and the regret bounded by variation methods [46].

**Quantum Computing and Learning.** Quantum computing has been rapidly advancing in recent years. Specifically, many machine learning problems are known to have significant quantum speedups, e.g., support vector machines [42], principal component analysis [34], classification [31, 33], etc. The combination of quantum computing and online learning has recently become a popular topic. For instance, online learning tools have been applied to solving semidefinite programs (SDPs) with quantum speedup in the problem dimension and the number of constraints [7, 50, 8, 49]. In addition, as an important quantum information task, the online version of quantum state learning has been systematically developed with good theoretical and empirical guarantees [1, 52, 14, 13].

For finding the Nash equilibrium of zero-sum games, a quantum algorithm was proposed in van Apeldoorn and Gilyén [48] by "quantizing" the classical algorithm in Grigoriadis and Khachiyan [22], achieving a quadratic speedup in the dimension parameters $m$ and $n$. At the same time, quantum algorithms for training linear and kernel-based classifiers were proposed in Li et al. [33], which have similar problem formulations to zero-sum games. Recently, an improved quantum algorithm for zero-sum games was proposed in Bouland et al. [6] using dynamic Gibbs sampling. All of the above quantum algorithms are based on the multiplicative weight update method, and as a consequence, they all share the $O(\sqrt{T})$ regret bound.

---

[2] The complexity given in Bouland et al. [6] is $\widetilde{O}(\sqrt{m+n}/\varepsilon^{2.5} + 1/\varepsilon^3)$, wherein the former term $\sqrt{m+n}/\varepsilon^{2.5}$ dominates the complexity. See Footnote 4 and Remark 4.4 for discussions.

[3] Here, we require that the input matrix $\mathbf{A}$ satisfies $\|\mathbf{A}\| \leq 1$, while other works require $|A_{i,j}| \leq 1$.

Table 1: Online Algorithms for $\varepsilon$-Approximate Nash Equilibria of Zero-Sum Games.

| Approach | Type | Regret | Update Cost Per Round | Classical/Quantum Time Complexity |
|---|---|---|---|---|
| [22] | Classical | $\widetilde{O}(\sqrt{T})$ | $\widetilde{O}(m+n)$ | $\widetilde{O}((m+n)/\varepsilon^2)$ |
| [48] | Quantum | $\widetilde{O}(\sqrt{T})$ | $\widetilde{O}(\sqrt{m+n}/\varepsilon)$ | $\widetilde{O}(\sqrt{m+n}/\varepsilon^3)$ |
| [6] | Quantum | $\widetilde{O}(\sqrt{T})$ | $\widetilde{O}(\sqrt{m+n}/\varepsilon^{0.5})$ | $\widetilde{O}(\sqrt{m+n}/\varepsilon^{2.5})$ [2] |
| [46] | Classical | $\widetilde{O}(1)$ | $\widetilde{O}(mn)$ | $\widetilde{O}(mn/\varepsilon)$ |
| [10] | Classical | $\widetilde{O}(1)$ | $\widetilde{O}(\sqrt{mn(m+n)})$ | $\widetilde{O}(mn + \sqrt{mn(m+n)}/\varepsilon)$ |
| Our result [3] | Quantum | $\widetilde{O}(1)$ | $\widetilde{O}(\sqrt{m+n}/\varepsilon^{1.5})$ | $\widetilde{O}(\sqrt{m+n}/\varepsilon^{2.5})$ |

## 1.1 Main Result

Our result in this paper establishes a positive answer to the following open question: *Does there exist a learning algorithm with $\widetilde{O}(1)$ regret allowing quantum speedups?*

Inspired by the optimistic follow-the-regularized-leader algorithm proposed in Syrgkanis et al. [46], we propose a sample-based quantum online learning algorithm for zero-sum games with $O(\log(mn))$ total regret, which is near-optimal. If we run this algorithm for $T$ rounds, it will compute an $\widetilde{O}(1/T)$-approximate Nash equilibrium with high probability, achieving a quadratic speedup in dimension parameters $m$ and $n$. Formally, we have the following quantum online learning algorithm:

**Theorem 1.1** (Online learning for zero-sum games). *Suppose $T \leq \widetilde{O}(m+n)$. There is a quantum online algorithm for zero-sum game $\mathbf{A} \in \mathbb{R}^{m \times n}$ with $\|\mathbf{A}\| \leq 1$ such that it achieves a total regret of $O(\log(mn))$ with high probability after $T$ rounds, while each round takes quantum time $\widetilde{O}(T^{1.5}\sqrt{m+n})$.*

Our algorithm does not need to read all the entries of the input matrix $\mathbf{A}$ at once. Instead, we assume that our algorithm can query its entries when necessary. The input model is described as follows:

- Classically, given any $i \in [m], j \in [n]$, the entry $A_{i,j}$ can be accessed in $\widetilde{O}(1)$ time.

- Quantumly, we assume that the entry $A_{i,j}$ can be accessed in $\widetilde{O}(1)$ time *coherently*.

This is the *standard quantum input model* for zero-sum games adopted in previous literature [33, 48, 6]. See more details in Section 2.4.

In addition, same as prior works [48, 6], our algorithm outputs *purely classical vectors* with succinct descriptions because they are sparse (with at most $T^2$ nonzero entries). Overall, using standard quantum inputs and generating classical outputs significantly facilitate end-to-end applications of our algorithm in the near term.

As a direct corollary, we can find an *$\varepsilon$-approximate Nash equilibrium* by taking $T = \widetilde{O}(1/\varepsilon)$, resulting in a quantum speedup stated as follows. A detailed comparison to previous literature is presented in Table 1.

**Corollary 1.2** (Computing Nash equilibrium). *There is a quantum online algorithm for zero-sum game $\mathbf{A} \in \mathbb{R}^{m \times n}$ with $\|A\| \leq 1$ that, with high probability, computes an $\varepsilon$-approximate Nash equilibrium in quantum time $\widetilde{O}(\sqrt{m+n}/\varepsilon^{2.5})$.[4]*

**Quantum Lower Bounds.** In the full version of [33], they showed a lower bound $\Omega(\sqrt{m+n})$ for the quantum query complexity of computing an $\varepsilon$-approximate Nash equilibrium of zero-sum games for constant $\varepsilon$. Therefore, our algorithm is tight in terms of $m$ and $n$.

---

[4]In fact, a condition of $\varepsilon = \Omega((m+n)^{-1})$ is required in our quantum algorithm. Nevertheless, our claim still holds because when $\varepsilon = O((m+n)^{-1})$, we can directly apply the classical algorithm in Grigoriadis and Khachiyan [22] with time complexity $\widetilde{O}((m+n)/\varepsilon^2) \leq \widetilde{O}(\sqrt{m+n}/\varepsilon^{2.5})$.

## 1.2  Our Techniques

Our quantum online algorithm is a stochastic modification of the optimistic multiplicative weight update proposed by Syrgkanis et al. [46]. We choose to "quantize" the optimistic online algorithm because it has a better convergence rate for zero-sum games than general multiplicative weight update algorithms. During the update of classical online algorithms, the update term (gradient vector) is computed in linear time by arithmetic operations. However, we observe that it is not necessary to know the exact gradient vector. This motivates us to apply stochastic gradient methods for updates so that certain fast quantum samplers can be utilized here and bring quantum speedups.

Specifically, in our quantum online algorithm, we need to establish an upper bound on the expectation of the total regret of our stochastic update rule, and also deal with errors that may appear from noisy samplers. To reduce the variance of the gradient estimator, we need multiple samples (from Gibbs distributions) at a time. To this end, we develop a fast quantum multi-Gibbs sampler that produces multiple samples by preparing and measuring quantum states.

**Sample-Based Optimistic Multiplicative Weight Update.**  Optimistic online learning adds a "prediction loss" term to the cumulative loss for regularized minimization, giving a faster convergence rate than the non-optimistic versions for zero-sum games. Arora et al. [4] surveyed the use of the multiplicative weight update method in various domains, but little was known for the optimistic learning method at that time. Daskalakis et al. [17] proposed an extragradient method that largely resembles the optimistic multiplicative weight. Syrgkanis et al. [46] gave a characterization of this update rule—RVU (Regret bounded by Variation in Utilities) property, which is very useful in proving regret bounds. Subsequently, the optimistic learning method is applied to other areas, including training GANs [18] and multi-agent learning [40].

However, when implementing the optimistic learning methods, we face a fundamental difficulty—we cannot directly access data from quantum states without measurement. To resolve this issue, we get samples from the desired distribution and use them to estimate the actual gradient. This idea is commonly seen in previous literature on quantum SDP solvers [7, 50, 8, 49]. Then we prove the regret bound (see Theorem 3.2) of our algorithm by showing that it has a property similar to the RVU property [46]. Moreover, we need multiple samples to obtain a small "variance" of the stochastic gradient (by taking the average of the samples), to ensure that the expected regret is bounded. Our fast quantum multi-Gibbs sampler produces the required samples and ensures further quantum speedups. In a nutshell, we give an algorithm (Algorithm 1) which modifies the optimistic multiplicative weight algorithm in Syrgkanis et al. [46] to fit the quantum implementation. This is *the first quantum algorithm that implements optimistic online learning* to the best of our knowledge.

**Fast Quantum Multi-Gibbs Sampling.**  The key to our sample-based approach is to obtain multiple samples from the Gibbs distribution after a common preprocessing step. For a vector $p \in \mathbb{R}^n$ with $\max_{i \in [n]} |p_i| \leq \beta$, a sample from the Gibbs distribution with respect to $p$ is a random variable $j \in [n]$ such that $\mathbf{Pr}\,[j = l] = \frac{\exp(p_l)}{\sum_{i=1}^n \exp(p_i)}$. Gibbs sampling on a quantum computer was first studied in Poulin and Wocjan [41], and was later used as a subroutine in quantum SDP solvers [7, 50, 8, 49]. However, the aforementioned quantum Gibbs samplers produce one sample from an $n$-dimensional Gibbs distribution in quantum time $\widetilde{O}(\beta\sqrt{n})$; thus, we can produce $k$ samples in quantum time $\widetilde{O}(\beta k\sqrt{n})$. Inspired by the recent work Hamoudi [26] about preparing multiple samples of quantum states, we develop a fast quantum Gibbs sampler (Theorem 4.2) which *produces $k$ samples from a Gibbs distribution in quantum time $\widetilde{O}(\beta\sqrt{nk})$*. Our quantum multi-Gibbs sampling may have potential applications in sample-based approaches for optimization tasks that require multiple samples.

Technically, the main idea is based on quantum rejection sampling [24, 39], where the target quantum state $|u\rangle$ is obtained by post-selection from a quantum state $|u_{\text{guess}}\rangle$ that is easy to prepare (see Section 4). The algorithm has the following steps (Here we assume $\beta = O(1)$ for simplicity): To bring $|u_{\text{guess}}\rangle$ closer to the target $|u\rangle$, we find the $k$ (approximately) most dominant amplitudes of $|u\rangle$ by quantum $k$-maximum finding [19] in quantum time $\widetilde{O}(\sqrt{nk})$. In quantum $k$-maximum finding, we need to estimate the amplitudes of $|u\rangle$ and compare them coherently, which requires the estimation should be consistent. To address this issue, we develop consistent quantum amplitude estimation (see Appendix C) for our purpose based on consistent phase estimation [2, 47], which is of independent interest. Then, we correct the tail amplitudes of $|u_{\text{guess}}\rangle$ by quantum singular

value transformation [20], resulting in an (approximate) target state with amplitude $\Omega(\sqrt{k/n})$ (see Appendix D for details). Finally, we post-select the target state by quantum amplitude amplification [9] in quantum time $\widetilde{O}(\sqrt{n/k})$. This follows that $k$ samples of the target quantum state can be obtained in $k \cdot \widetilde{O}(\sqrt{n/k}) = \widetilde{O}(\sqrt{nk})$ time.

We believe that our technique can be extended to a wide range of distributions whose mass function is monotonic and satisfies certain Lipschitz conditions.

## 2 Preliminaries

### 2.1 General Mathematical Notations

For convenience, we use $[n]$ to denote the set $\{1, 2, \ldots, n\}$. We use $e_i$ to denote a vector whose $i$-th coordinate is 1 and other coordinates are 0. For a vector $v \in \mathbb{R}^n$, $v_i$ is the $i$-th coordinate of $v$. For a function $f : \mathbb{R} \to \mathbb{R}$, we write $f(v)$ to denote the result of $f$ applied to its coordinates, i.e., $f(v) = (f(v_1), f(v_2), \ldots, f(v_n))$. We use $\Delta_n$ to represent the set of $n$-dimensional probability distributions, i.e., $\Delta_n := \{v \in \mathbb{R}^n : \sum_{i=1}^n v_i = 1, \forall i \in [n], v_i \geq 0\}$. Here the $i$-th coordinate represents the probability of event $i$ takes place. We use $\|\cdot\|$ for vector norms. The $l_1$ norm $\|\cdot\|_1$ for a vector $v \in \mathbb{R}^n$ is defined as $\|v\|_1 := \sum_{i=1}^n |v_i|$. For two $n$-dimensional probability distributions $p, q \in \Delta_n$, their total variance distance is defined as: $d_{\text{TV}}(p, q) = \frac{1}{2}\|p - q\|_1 = \frac{1}{2}\sum_{i=1}^n |p_i - q_i|$. We will use $\mathbf{A} \in \mathbb{R}^{m \times n}$ to denote a matrix with $m$ rows and $n$ columns. $A_{i,j}$ is the entry of $\mathbf{A}$ in the $i$-th row and $j$-th column. $\|\mathbf{A}\|$ denotes the operator norm of matrix $\mathbf{A}$. For a vector $v \in \mathbb{R}^n$, we use $\text{diag}(v)$ to denote the diagonal matrix in $\mathbb{R}^{n \times n}$ whose diagonal entries are coordinates of $v$ with the same order.

### 2.2 Game Theory

Let us consider two players, Alice and Bob, playing a zero-sum game represented by a matrix $\mathbf{A} \in \mathbb{R}^{m \times n}$ with entries $A_{i,j}$, where $[m]$ is the labeled action set of Alice and $[n]$ of Bob. Usually, Alice is called the row player and Bob is called the column player. $A_{i,j}$ is the payoff to Bob and $-A_{i,j}$ is the payoff to Alice when Alice plays action $i$ and Bob plays action $j$. Both players want to maximize their payoff when they consider their opponent's strategy.

Consider the situation that both players' actions are the best responses to each other. In this case, we call the actions form a *Nash equilibrium*. A famous minimax theorem by von Neumann [51] states that we can exchange the order of the min and max operation and thus the value of the Nash equilibrium of the game can be properly defined. To be more exact, we have: $\min_{x \in \Delta_m} \max_{y \in \Delta_n} x^\mathsf{T} \mathbf{A} y = \max_{y \in \Delta_n} \min_{x \in \Delta_m} x^\mathsf{T} \mathbf{A} y$. Here the value of the minimax optimization problem is called the *value of the game*. Our goal is to find an approximate Nash equilibrium for this problem. More formally, we need two probabilistic vectors $x \in \Delta_m, y \in \Delta_n$ such that the following holds: $\max_{y' \in \Delta_n} x^\mathsf{T} \mathbf{A} y' - \min_{x' \in \Delta_m} x'^\mathsf{T} \mathbf{A} y \leq \varepsilon$. We will call such a pair $(x, y)$ an $\varepsilon$-*approximate Nash Equilibrium* for the two-player zero-sum game $\mathbf{A}$. Computing an $\varepsilon$-approximate Nash for a zero-sum game is not as hard as for a general game. For a general game, approximately computing its Nash equilibrium is PPAD-hard [16, 12].

Here, we emphasize an important observation that will be used throughout our paper: we can add a constant or multiply by a positive constant on all $\mathbf{A}$'s entries without changing the solution pair of the Nash equilibrium. This is because, in the definition of the Nash equilibrium, the best response is always in comparison with other possible actions, so only the order of the utility matters. Thus without loss of generality, we can always rescale $\mathbf{A}$ to $(\mathbf{A} + c\mathbf{1})/2$ where $\mathbf{1}$ is an $m \times n$ matrix with all entries being 1 with $c$ being the largest absolute value of $\mathbf{A}$'s entries, and let $\mathbf{A}' = \mathbf{A}/\|\mathbf{A}\|$ to guarantee that $\mathbf{A}$ has non-negative entries and has operator norm no more than 1.

### 2.3 Online Learning

#### 2.3.1 Notions of online learning

In general, online learning focuses on making decisions in an uncertain situation, where a decider is not aware of the current loss and is required to make decisions based on observations of previous

losses. To be more exact, we fix the number of rounds $T$ and judge the performance of the algorithm (in the following of this subsection we use "decider" with the same meaning as the "algorithm") in these $T$ rounds. Assume that the decider is allowed to choose actions in the domain $\mathcal{X}$, usually a convex subset of a finite-dimensional Euclidean space. Let $t \in [T]$ denote the current number of rounds. At round $t$, the decider chooses an action $x_t \in \mathcal{X}$. (The action may depend on the decider's previous observation of the loss functions $l_i$ for all $i \in [t-1]$.) Then the decider will get the loss $l_t(x_t)$ for the current round. We assume that the decider can also observe the full information of $l_t$, i.e., the formula of the function. We judge the performance of the algorithm by its *regret*: $\mathcal{R}(T) = \sum_{i=1}^{T} l_t(x_t) - \min_{x \in \mathcal{X}} \sum_{i=1}^{T} l_t(x)$. Intuitively, the regret shows how far the total loss in $T$ rounds caused by the algorithm is from the optimal static loss.

### 2.3.2 Online learning in zero-sum games

To demonstrate how to compute the approximate Nash equilibrium using online learning algorithms, we present a useful proposition here. It states that from any sublinear regret learning algorithm $\mathcal{A}$ with regret $\mathcal{R}(T)$, we can find an $O(\mathcal{R}(T)/T)$-approximate Nash equilibrium of the zero-sum game in $T$ rounds.

To be more precise, let us consider the following procedure. Let $\mathbf{A}$ be the matrix for the two-player zero-sum game. The algorithm starts with some initial strategies $u_0 \in \Delta_m, v_0 \in \Delta_n$ for the two players. Then at each round $t$, for each player, it makes decisions with previous observations of the opponent's strategy. In particular, the row player is required to choose his/her action $u_t \in \Delta_m$ after considering the previous loss functions $g_i(x) = v_i^\mathsf{T} \mathbf{A}^\mathsf{T} x$ for $i \in [t-1]$. Similarly, the column player chooses his/her action $v_t \in \Delta_n$ with respect to the previous loss functions $h_i(y) = -u_i^\mathsf{T} \mathbf{A} y$ for $i \in [t-1]$. After both players choose their actions at this round $t$, they will receive their loss functions $g_t(x) := v_t^\mathsf{T} \mathbf{A}^\mathsf{T} x$ and $h_t(y) = -u_t^\mathsf{T} \mathbf{A} y$, respectively.

Suppose after $T$ rounds, the regret of the row player with respect to the loss functions $g_t(x)$ is $\mathcal{R}(T)$, and the regret of the column player with loss functions $h_t(y)$ is $\mathcal{R}'(T)$. We can write the total regret $\mathcal{R}(T) + \mathcal{R}'(T)$ explicitly: $\mathcal{R}(T) + \mathcal{R}'(T) = T(\max_{v \in \Delta_n} \langle v, \mathbf{A}^\mathsf{T} \hat{u} \rangle - \min_{u \in \Delta_m} \langle u, \mathbf{A} \hat{v} \rangle)$, where the average strategy is defined as $\hat{u} = \sum_{i=1}^{T} u_i / T, \hat{v} = \sum_{i=1}^{T} v_i / T$. This pair is a good approximation of the Nash equilibrium for the game $\mathbf{A}$ if the regret is $o(T)$.

### 2.4 Quantum Computing

In quantum mechanics, a $d$-dimensional quantum state is described by a unit vector $v = (v_0, v_1, \ldots, v_{d-1})^\mathsf{T}$, usually denoted as $|v\rangle$ with the Dirac symbol $|\cdot\rangle$, in a complex Hilbert space $\mathbb{C}^d$. The computational basis of $\mathbb{C}^d$ is defined as $\{|i\rangle\}_{i=0}^{d-1}$, where $|i\rangle = (0, \ldots, 0, 1, 0, \ldots, 0)^\mathsf{T}$ with the $i$-th (0-indexed) entry being 1 and other entries being 0. The inner product of quantum states $|v\rangle$ and $|w\rangle$ is defined by $\langle v|w\rangle = \sum_{i=0}^{d-1} v_i^* w_i$, where $z^*$ denotes the conjugate of complex number $z$. The norm of $|v\rangle$ is defined by $\||v\rangle\| = \sqrt{\langle v|v\rangle}$. The tensor product of quantum states $|v\rangle \in \mathbb{C}^{d_1}$ and $|w\rangle \in \mathbb{C}^{d_2}$ is defined by $|v\rangle \otimes |w\rangle = (v_0 w_0, v_0 w_1, \ldots, v_{d_1-1} w_{d_2-1})^\mathsf{T} \in \mathbb{C}^{d_1 d_2}$, denoted as $|v\rangle|w\rangle$ for short.

A quantum bit (qubit for short) is a quantum state $|\psi\rangle$ in $\mathbb{C}^2$, which can be written as $|\psi\rangle = \alpha|0\rangle + \beta|1\rangle$ with $|\alpha|^2 + |\beta|^2 = 1$. An $n$-qubit quantum state is in the tensor product space of $n$ Hilbert spaces $\mathbb{C}^2$, i.e., $(\mathbb{C}^2)^{\otimes n} = \mathbb{C}^{2^n}$ with the computational basis $\{|0\rangle, |1\rangle, \ldots, |2^n - 1\rangle\}$. To obtain classical information from an $n$-qubit quantum state $|v\rangle$, we measure $|v\rangle$ on the computational basis and obtain outcome $i$ with probability $p(i) = |\langle i|v\rangle|^2$ for every $0 \leq i < 2^n$. The evolution of a quantum state $|v\rangle$ is described by a unitary transformation $U: |v\rangle \mapsto U|v\rangle$ such that $UU^\dagger = U^\dagger U = I$, where $U^\dagger$ is the Hermitian conjugate of $U$, and $I$ is the identity operator. A quantum gate is a unitary transformation that acts on 1 or 2 qubits, and a quantum circuit is a sequence of quantum gates.

Throughout this paper, we assume the quantum oracle $\mathcal{O}_\mathbf{A}$ for a matrix $\mathbf{A} \in \mathbb{R}^{m \times n}$, which is a unitary operator such that for every row index $i \in [m]$ and column index $j \in [n]$, $\mathcal{O}_\mathbf{A}|i\rangle|j\rangle|0\rangle = |i\rangle|j\rangle|A_{i,j}\rangle$. Intuitively, the oracle $\mathcal{O}_\mathbf{A}$ reads the entry $A_{i,j}$ and stores it in the third register; this is potentially stronger than the classical counterpart when the query is a linear combination of basis vectors, e.g., $\sum_k \alpha_k |i_k\rangle|j_k\rangle$ with $\sum_k |\alpha_k|^2 = 1$. This is known as the *superposition* principle in quantum computing.

Note that this input model for matrices is commonly used in quantum algorithms, e.g., linear system solvers [28] and semidefinite programming solvers [7, 50, 8, 49].

A quantum (query) algorithm $\mathcal{A}$ is a quantum circuit that consists a sequence of unitary operators $G_1, G_2, \ldots, G_T$, each of which is either a quantum gate or a quantum oracle. The quantum time complexity of $\mathcal{A}$ is measured by the number $T$ of quantum gates and quantum oracles in $\mathcal{A}$. The execution of $\mathcal{A}$ on $n$ qubits starts with quantum state $|0\rangle^{\otimes n}$, then it performs unitary operators $G_1, G_2, \ldots, G_T$ on the quantum state in this order, resulting in the quantum state $|\phi\rangle = G_T \ldots G_2 G_1 |0\rangle^{\otimes n}$. Finally, we measure $|\phi\rangle$ on the computational basis $|i\rangle$ for $0 \le i < 2^n$, giving a classical output $i$ with probability $|\langle i|\phi\rangle|^2$.

## 3 Quantum Algorithm for Online Zero-Sum Games by Sample-Based Optimistic Multiplicative Weight Update

Now, we present our quantum algorithm for finding an approximate Nash equilibrium for zero-sum games (Algorithm 1). This algorithm is a modification of the optimistic multiplicative weight algorithm [46], in which we use stochastic gradients to estimate true gradients. This modification utilizes the quantum advantage of Gibbs sampling (as will be shown in Section 4). It is the source of quantum speedups in the algorithm and also the reason that we call the algorithm *sample-based*. To this end, we first give the definition of Gibbs sampling oracles.

**Definition 3.1** (Approximate Gibbs sampling oracle). Let $p \in \mathbb{R}^n$ be an $n$-dimensional vector, $\epsilon > 0$ be the approximate error. We let $\mathcal{O}_p^{\text{Gibbs}}(k, \epsilon)$ denote the oracle which produces $k$ independent samples from a distribution that is $\epsilon$-close to the Gibbs distribution with parameter $p$ in total variation distance. Here, for a random variable $j$ taking value in $[n]$ following the Gibbs distribution with parameter $p$, we have $\mathbf{Pr}\,[j = l] = \exp(p_l)/\sum_{i=1}^n \exp(p_i)$.

---

**Algorithm 1** Sample-Based Optimistic Multiplicative Weight Update for Matrix Games

---

**Input:** $\mathbf{A} \in \mathbb{R}^{m \times n}$, additive approximation $\varepsilon$, approximate Gibbs sampling oracle $\mathcal{O}^{\text{Gibbs}}$ with error $\epsilon_{\text{G}}$, total episode $T$, learning rate $\lambda \in (0, \sqrt{3}/6)$.
**Output:** $(\hat{u}, \hat{v})$ as the approximate Nash equilibrium of the matrix game $\mathbf{A}$.
 1: Initialize $\hat{u} \leftarrow \mathbf{0}_m, \hat{v} \leftarrow \mathbf{0}_n, x_1 \leftarrow \mathbf{0}_m, y_1 \leftarrow \mathbf{0}_n$.
 2: Set $g_1 \leftarrow x_1, h_1 \leftarrow y_1$.
 3: **for** $t = 1, \ldots, T$ **do**
 4:      Get $T$ independent samples $i_1^t, i_2^t, \ldots, i_T^t$ from the Gibbs sampling oracle $\mathcal{O}_{-\mathbf{A}h_t}^{\text{Gibbs}}(T, \epsilon_{\text{G}})$.
 5:      Choose the action $\zeta_t = \sum_{N=1}^T e_{i_N^t}/T$.
 6:      Update $x_{t+1} \leftarrow x_t + \lambda \zeta_t, g_{t+1} \leftarrow x_{t+1} + \lambda \zeta_t$. $\hat{u} \leftarrow \hat{u} + \frac{1}{T}\zeta_t$.
 7:      Get $T$ independent samples $j_1^t, j_2^t, \ldots, j_T^t$ from the Gibbs sampling oracle $\mathcal{O}_{\mathbf{A}^\mathsf{T} g_t}^{\text{Gibbs}}(T, \epsilon_{\text{G}})$.
 8:      Choose the action $\eta_t = \sum_{N=1}^T e_{j_N^t}/T$.
 9:      Update $y_{t+1} \leftarrow y_t + \lambda \eta_t, h_{t+1} \leftarrow y_{t+1} + \lambda \eta_t$, $\hat{v} \leftarrow \hat{v} + \frac{1}{T}\eta_t$.
10: **end for**
11: **return** $(\hat{u}, \hat{v})$.

---

Suppose the zero-sum game is represented by matrix $\mathbf{A} \in \mathbb{R}^{m \times n}$ with $\|\mathbf{A}\| \le 1$. Our sample-based optimistic multiplicative weight update algorithm is given in Algorithm 1. Algorithm 1 is inspired by the classical optimistic follow-the-regularized-leader algorithm (see Appendix A for more information). In that classical algorithm, the update terms are essentially $\mathbb{E}[\zeta_t]$ and $\mathbb{E}[\eta_t]$, which are computed deterministically by matrix arithmetic operations during the update. In contrast, we do this probabilistically by sampling from the Gibbs distributions and $\zeta_t$ and $\eta_t$ in Line 5 and Line 10 are the corresponding averages of the samples. For this to work, we need to bound the expectation of the total regret (see Appendix B) based on the RVU property (Definition A.2). Technically, the $l_1$ variances of $\zeta_t$ and $\eta_t$ turn out to be significant in the analysis. To reduce the variances, we need multiple independent samples identically distributed from Gibbs distributions (see Line 4 and Line 7 in Algorithm 1). Because of the randomness from Gibbs sampling oracles, the total regret $\mathcal{R}(T) + \mathcal{R}'(T)$ of Algorithm 1 is a random variable. Nevertheless, we can bound the total regret by $O(\log(mn))$ with high probability as follows.

**Theorem 3.2.** *Let $\epsilon_G = 1/T$. After $T$ rounds of playing the zero-sum game $\mathbf{A}$ by Algorithm 1, the total regret will be bounded by*

$$\mathcal{R}(T) + \mathcal{R}'(T) \leq 144\lambda + \frac{3\log(mn)}{\lambda} + 12,$$

*with probability at least $2/3$. Then, for any constant $\lambda \in (0, \sqrt{3}/6)$, the total regret is $O(\log(mn))$. Moreover, if we choose $T = \widetilde{\Theta}(1/\varepsilon)$, then Algorithm 1 will return an $\varepsilon$-approximate Nash equilibrium of the game $\mathbf{A}$.*

The proof of Theorem 3.2 is deferred to Appendix B.2. Combining Theorem 3.2 and our fast quantum multi-Gibbs sampler in Theorem 4.2 (which will be developed in the next section), we obtain a quantum algorithm for finding an $\varepsilon$-approximate Nash equilibrium of zero-sum games.

**Corollary 3.3.** *If we choose $T = \widetilde{\Theta}(1/\varepsilon)$ and use quantum multi-Gibbs sampling (Algorithm 2) with $\epsilon_G = 1/T$, then Algorithm 1 will return an $\varepsilon$-approximate Nash equilibrium in quantum time $\widetilde{O}(\sqrt{m+n}/\varepsilon^{2.5})$.*

## 4 Fast Quantum Multi-Gibbs Sampling

In Algorithm 1, the vectors $h_t$ and $g_t$ updated in each round are used to generate independent samples from Gibbs distributions $\mathcal{O}^{\text{Gibbs}}_{-\mathbf{A}h_t}$ and $\mathcal{O}^{\text{Gibbs}}_{\mathbf{A}^\top g_t}$. Here, $h_t$ and $g_t$ are supposed to be stored in classical registers. To allow quantum speedups, we store $h_t$ and $g_t$ in quantum-read classical-write random access memory (QRAM) [21], which is commonly used in prior work [49, 6]. Specifically, a QRAM can store/modify an array $a_1, a_2, \ldots, a_\ell$ of classical data and provide quantum (read-only) access to them, i.e., a unitary operator $U_{\text{QRAM}}$ is given such that $U_{\text{QRAM}}: |i\rangle|0\rangle \mapsto |i\rangle|a_i\rangle$. Without loss of generality (see Remark 4.3), suppose we have quantum oracle $\mathcal{O}_{\mathbf{A}}$ for $\mathbf{A} \in \mathbb{R}^{n \times n}$ with $A_{i,j} \in [0,1]$, and QRAM access to a vector $z \in \mathbb{R}^n$ with $z_i \geq 0$. We also need the polynomial approximation of the exponential function for applying the QSVT technique [20]:

**Lemma 4.1** (Polynomial approximation, Lemma 7 of [48]). *Let $\beta \geq 1$ and $\epsilon_P \in (0, 1/2)$. There is a classically efficiently computable polynomial $P_\beta \in \mathbb{R}[x]$ of degree $O(\beta \log(\epsilon_P^{-1}))$ such that $\left| P_\beta(x) \right| \leq 1$ for $x \in [-1,1]$, and $\max_{x \in [-1,0]} \left| P_\beta(x) - \frac{1}{4}\exp(\beta x) \right| \leq \epsilon_P$.*

Then, we can produce multiple samples from $\mathcal{O}^{\text{Gibbs}}_{\mathbf{A}z}$ efficiently by Algorithm 2 on a quantum computer. Algorithm 2 is inspired by Hamoudi [26] about preparing multiple samples of a quantum state, with quantum access to its amplitudes. However, we do not have access to the exact values of the amplitudes, which are $(\mathbf{A}z)_i$ in our case. To resolve this issue, we develop consistent quantum amplitude estimation (see Appendix C) to estimate $(\mathbf{A}z)_i$ with a unique answer (Line 1). After having prepared an initial quantum state $|u_{\text{guess}}\rangle$, we use quantum singular value decomposition [20] to correct the tail amplitudes (Line 6), and finally obtain the desired quantum state $|\tilde{u}_{\text{Gibbs}}\rangle$ by quantum amplitude amplification [9] (Line 7). We have the following (see Appendix D for its proof):

**Theorem 4.2** (Fast quantum multi-Gibbs sampling). *For $k \in [n]$, if we set $\epsilon_P = \Theta(k\epsilon_G^2/n)$, then Algorithm 2 will produce $k$ independent and identical distributed samples from a distribution that is $\epsilon_G$-close to $\mathcal{O}^{\text{Gibbs}}_{\mathbf{A}z}$ in total variation distance, in quantum time $\widetilde{O}(\beta\sqrt{nk})$.*

*Remark* 4.3. If $\mathbf{A} \in \mathbb{R}^{m \times n}$ is not a square matrix, then by adding 0's we can always enlarge $\mathbf{A}$ to an $(m+n)$-dimensional square matrix. For $\mathcal{O}^{\text{Gibbs}}_{-\mathbf{A}h_t}$ as required in Algorithm 1, we note that $\mathcal{O}^{\text{Gibbs}}_{(\mathbf{1}-\mathbf{A})h_t}$ indicates the same distribution as $\mathcal{O}^{\text{Gibbs}}_{-\mathbf{A}h_t}$, where $\mathbf{1}$ has the same size as $\mathbf{A}$ with all entries being 1. From the above discussion, we can always convert $\mathbf{A}$ to another matrix satisfying our assumption, i.e., with entries in the range $[0,1]$.

*Remark* 4.4. The description of the unitary operator $U^{\text{exp}}$ defined by the polynomial $P_{2\beta}$ can be classically computed to precision $\epsilon_P$ in time $\widetilde{O}(\beta^3)$ by Haah [25], which is $\widetilde{O}(1/\varepsilon^3)$ in our case as $\beta \leq \lambda T = \widetilde{\Theta}(1/\varepsilon)$ is required in Corollary 3.3. This extra cost can be neglected because $\widetilde{O}(\sqrt{m+n}/\varepsilon^{2.5})$ dominates the complexity whenever $\varepsilon = \Omega((m+n)^{-1})$.

*Remark* 4.5. Our multi-Gibbs sampler is based on maximum finding and consistent amplitude estimation, with a guaranteed worst-case performance in each round. In comparison, the dynamic Gibbs sampler in [6] maintains a hint vector, resulting in an amortized time complexity per round.

**Algorithm 2** Quantum Multi-Gibbs Sampling $\mathcal{O}_{\mathbf{A}z}^{\text{Gibbs}}(k, \epsilon_{\text{G}})$

---

**Input:** Quantum oracle $\mathcal{O}_{\mathbf{A}}$ for $\mathbf{A} \in \mathbb{R}^{n \times n}$, QRAM access to $z \in \mathbb{R}^n$ with $\|z\|_1 \leq \beta$, polynomial $P_{2\beta}$ with parameters $\epsilon_P$ by Lemma 4.1, number $k$ of samples. We write $u = \mathbf{A}z$.

**Output:** $k$ independent samples $i_1, i_2, \ldots, i_k$.

1: Obtain $\mathcal{O}_{\tilde{u}} \colon |i\rangle|0\rangle \mapsto |i\rangle|\tilde{u}_i\rangle$ by consistent quantum amplitude estimation such that $u_i \leq \tilde{u}_i \leq u_i + 1$.

2: Find the set $S \subseteq [n]$ of indexes of the $k$ largest $\tilde{u}_i$ by quantum $k$-maximum finding, with access to $\mathcal{O}_{\tilde{u}}$.

3: Obtain $\tilde{u}_i$ for all $i \in S$ from $\mathcal{O}_{\tilde{u}}$, then compute $\tilde{u}^* = \min\limits_{i \in S} \tilde{u}_i$ and $W = \sum\limits_{i \in S} \exp(\tilde{u}_i) + (n - k)\exp(\tilde{u}^*)$.

4: **for** $\ell = 1, \ldots, k$ **do**

5:     Prepare the quantum state $|u_{\text{guess}}\rangle = \sum_{i \in S} \sqrt{\frac{\exp(\tilde{u}_i)}{W}} |i\rangle + \sum_{i \notin S} \sqrt{\frac{\exp(\tilde{u}^*)}{W}} |i\rangle$.

6:     Obtain unitary $U^{\text{exp}}$ such that $\langle 0|^{\otimes a} U^{\text{exp}} |0\rangle^{\otimes a} = \text{diag}\big(P_{2\beta}(u - \max\{\tilde{u}, \tilde{u}^*\})\big)/4\beta$ by QSVT.

7:     Post-select $|\tilde{u}_{\text{Gibbs}}\rangle \propto \langle 0|^{\otimes a} U^{\text{exp}} |u_{\text{guess}}\rangle |0\rangle^{\otimes a}$ by quantum amplitude amplification.

8:     Let $i_\ell$ be the measurement outcome of $|\tilde{u}_{\text{Gibbs}}\rangle$ in the computational basis.

9: **end for**

10: **return** $i_1, i_2, \ldots, i_k$.

---

## 5 Discussion

In our paper, we propose the first quantum online algorithm for zero-sum games with near-optimal regret. This is achieved by developing a sample-based stochastic version of the optimistic multiplicative weight update method [46]. Our core technical contribution is a fast multi-Gibbs sampling, which may have potential applications in other quantum computing scenarios.

Our result naturally gives rise to some further open questions. For instance: Can we improve the dependence on $\varepsilon$ for the time complexity? And can we further explore the combination of optimistic learning and quantum computing into broader applications? Now that many heuristic quantum approaches for different machine learning problems have been realized, e.g.in Havlíček et al. [29], Saggio et al. [44], Harrigan et al. [27], can fast quantum algorithms for zero-sum games be realized in the near future?

## Acknowledgments

We would like to thank Kean Chen, Wang Fang, Ji Guan, Junyi Liu, Xinzhao Wang, Chenyi Zhang, and Zhicheng Zhang for helpful discussions. MG would like to thank Mingsheng Ying for valuable suggestions.

Minbo Gao was supported by the National Key R&D Program of China (2018YFA0306701) and the National Natural Science Foundation of China (61832015). Zhengfeng Ji was supported by a startup fund from Tsinghua University, and the Department of Computer Science and Technology, Tsinghua Univeristy. Tongyang Li was supported by a startup fund from Peking University, and the Advanced Institute of Information Technology, Peking University. Qisheng Wang was supported by the MEXT Quantum Leap Flagship Program (MEXT Q-LEAP) grants No. JPMXS0120319794.

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

## A  Revisit of Optimistic Online Learning

In this section, we briefly review some important properties in the classical optimistic online learning algorithms. Some of the propositions in this section will be frequently used in the proof of the regret bound.

For convenience, we will use $\psi(\cdot)$ to denote the negative entropy function, i.e., $\psi \colon \Delta_n \to \mathbb{R}$, $\psi(p) = \sum_{i=1}^n p_i \log p_i$. Note that log stands for the natural logarithm function with base e.

For a vector norm $\|\cdot\|$, its dual norm is defined as:
$$\|y\|_* = \max_x \left\{ \langle x, y \rangle : \|x\| \leq 1 \right\}.$$

**Proposition A.1.** *Let L be a vector in n-dimensional space. If $p^* = \arg\min\limits_{p \in \Delta_n} \left\{ \langle p, L \rangle + \psi(p) \right\}$, then*

*$p^*$ can be written as:*
$$p^* = \frac{\exp(-L)}{\|\exp(-L)\|_1}$$

*and vice versa.*

*Proof.* Write $L = (L_1, L_2, \ldots, L_n)$. By definition, we know that $p^*$ is the solution to the following convex optimization problem:

$$\begin{aligned}
\underset{p_1, p_2, \ldots, p_n}{\text{minimize}} \quad & \sum_{i=1}^n p_i L_i + \sum_{i=1}^n p_i \log p_i \\
\text{subject to} \quad & \sum_{i=1}^n p_i = 1, \\
& \forall i, p_i \geq 0
\end{aligned}$$

The Lagrangian is

$$\mathcal{L}(p, u, v) = \sum_{i=1}^n p_i L_i + \sum_{i=1}^n p_i \log p_i - \sum_{i=1}^n u_i p_i + v \left( \sum_{i=1}^n p_i - 1 \right)$$

From KKT conditions, we know that the stationarity is:
$$L_i + 1 + \log p_i - u_i + v = 0. \tag{1}$$

The complementary slackness is:
$$u_i p_i = 0.$$

The primal feasibility is
$$\forall i, p_i \geq 0; \sum_{i=1}^n p_i = 1.$$

The dual feasibility is
$$u_i \geq 0.$$

If $u_i \neq 0$ then $p_i = 0$, from stationarity we know $u_i = -\infty$, but that violates the dual feasibility. So we can conclude that $u_i = 0$ for all $i \in [n]$, thus $p_i \propto \exp(-L_i)$ and the result follows. $\square$

Now we present a generalized version of the optimistic multiplicative weight algorithm called optimisitically follow the regularized leader (Opt-FTRL) in Algorithm 3. In the algorithm, $m_t$ has the same meaning as $m(t)$ for notation consistency.

---

**Algorithm 3** Optimistic follow-the-regularized-leader

---

**Input:** The closed convex domain $\mathcal{X}$.
**Output:** Step size $\lambda$, loss gradient prediction $m$.
  Initialize $L_0 \leftarrow 0$, choose appropriate $m_1$.
  **for** $t = 1, \ldots, T$ **do**
    **Choose** $x_t = \arg\min\limits_{x \in \mathcal{X}} \left\{ \lambda \langle L_{t-1} + m_t, x \rangle + \psi(x) \right\}$.
    Observe loss $l_t$, update $L_t = L_{t-1} + l_t$.
    Compute $m_{t+1}$ using observations till now.
  **end for**

---

Now we study a crucial property that leads to the fast convergence of the algorithm, called the Regret bounded by Variation in Utilities (RVU in short). For simplicity, we only consider the linear loss function $l_t(x) = \langle l_t, x \rangle$. (There is a little abuse of notation here.)

**Definition A.2** (Regret bounded by Variations in Utilities (RVU), Definition 3 in Syrgkanis et al. [46]). Consider an online learning algorithm $\mathcal{A}$ with regret $\mathcal{R}(T) = o(T)$, we say that it has the property of regret bound by variation in utilities if for any linear loss sequence $l_1, l_2, \ldots, l_T$, there exists parameters $\alpha > 0, 0 < \beta \leq \gamma$ such that the algorithm output decisions $x_1, x_2, \ldots, x_T, x_{T+1}$ that satisfy:

$$\sum_{i=1}^{T} \langle l_i, x_i \rangle - \min_{x \in \mathcal{X}} \sum_{i=1}^{T} \langle l_i, x \rangle \leq \alpha + \beta \sum_{i=1}^{T-1} \|l_{i+1} - l_i\|_*^2 - \gamma \sum_{i=1}^{T-1} \|x_{i+1} - x_i\|^2,$$

where $\|\cdot\|_*$ is the dual norm of $\|\cdot\|$.

We do not choose the norm to be any specific one here. In fact, Syrgkanis et al. [46] have already shown that the above optimistic follow-the-regularized-leader algorithm has the RVU property with respect to any norm $\|\cdot\|$ in which the negative entropy function $\psi$ is 1-strongly convex. So, from Pinsker's inequality, for $l_2$ norms the following result holds:

**Proposition A.3** (Proposition 7 in Syrgkanis et al. [46]). *If we choose $m_t = l_{t-1}$ in the optimistic follow-the-regularized-leader algorithm with step size $\lambda \leq 1/2$, then it has the regret bound by variation in utilities property with the parameters $\alpha = \log n / \lambda$, $\beta = \lambda$ and $\gamma = 1/(4\lambda)$, where $n$ is the dimension of $\mathcal{X}$.*

# B  Regret Bound and Time Complexity of Our Algorithm

## B.1  Ideal Samplers

We assume that after the execution of our algorithm, the sequences we get are $\{(x_t, y_t)\}_{t=1}^{T+1}$ and $\{(g_t, h_t)\}_{t=1}^{T+1}$, respectively. We denote $u_t := \frac{\exp(-\mathbf{A}h_t)}{\|\exp(-\mathbf{A}h_t)\|_1}$ and $v_t := \frac{\exp(\mathbf{A}^\mathsf{T} g_t)}{\|\exp(\mathbf{A}^\mathsf{T} g_t)\|_1}$ to be the corresponding Gibbs distribution, we will first assume that the Gibbs oracle in our algorithm has no error (i.e. $\epsilon_G = 0$) until Theorem B.5 is proved.

**Observation B.1.** *The sequence $\{u_t\}_{t=1}^{T}$ can be seen as the decision result of applying optimistic FTRL algorithm to the linear loss function $\mathbf{A}\eta_t$ with linear prediction function $\mathbf{A}\eta_{t-1}$, and similarly for $\{v_t\}_{t=1}^{T+1}$ with the loss function $-\mathbf{A}^\mathsf{T}\zeta_t$, the prediction function $-\mathbf{A}^\mathsf{T}\zeta_{t-1}$.*

*Proof.* By symmetry, we only consider $u_t$. Since $u_t = \frac{\exp(-\mathbf{A}h_t)}{\|\exp(-\mathbf{A}h_t)\|_1}$, from Proposition A.1 we can write

$$u_t = \arg\min_{u \in \Delta_m} \left\{ \langle \mathbf{A}h_t, u \rangle + \psi(u) \right\}.$$

Then we notice the iteration of Algorithm 1 gives

$$h_t = \lambda \left( \sum_{i=1}^{t-1} \eta_i \right) + \lambda \eta_{t-1}.$$

So from the definition of the Algorithm 3, we know that our observation holds. $\square$

This observation, together with Proposition A.3, gives the following inequalities. For any $u \in \Delta_m$, $v \in \Delta_n$, we have:

$$\sum_{t=1}^{T} \langle u_t - u, \mathbf{A}\eta_t \rangle \leq \frac{\log m}{\lambda} + \lambda \sum_{t=1}^{T-1} \|\mathbf{A}(\eta_{t+1} - \eta_t)\|^2 - \frac{1}{4\lambda} \sum_{t=1}^{T-1} \|u_{t+1} - u_t\|^2, \quad (2)$$

$$\sum_{t=1}^{T} \langle v_t - v, -\mathbf{A}^\mathsf{T}\zeta_t \rangle \leq \frac{\log n}{\lambda} + \lambda \sum_{t=1}^{T-1} \|\mathbf{A}^\mathsf{T}(\zeta_{t+1} - \zeta_t)\|^2 - \frac{1}{4\lambda} \sum_{t=1}^{T-1} \|v_{t+1} - v_t\|^2. \quad (3)$$

However, we find that the loss function is slightly different from what we expect.

Let us consider the difference $q_t := \mathbf{A}(v_t - \eta_t)$ and $p_t := -\mathbf{A}^\mathsf{T}(u_t - \zeta_t)$, we have the decomposition of the regret:

$$\sum_{t=1}^{T} \langle u_t - u, \mathbf{A}v_t \rangle = \sum_{t=1}^{T} \langle u_t - u, \mathbf{A}\eta_t \rangle + \sum_{t=1}^{T} \langle u_t - u, q_t \rangle.$$

Notice that $\mathbb{E}[q_t] = \mathbb{E}[p_t] = 0$, we have:

**Lemma B.2.**

$$\mathbb{E}\left[\sum_{t=1}^{T} \langle u_t - u, q_t \rangle\right] = 0, \mathbb{E}\left[\sum_{t=1}^{T} \langle v_t - v, p_t \rangle\right] = 0$$

*Proof.* By symmetry, we only prove the case for $u$. It suffices to prove that for every $t$, $\mathbb{E}[\langle u_t - u, q_t \rangle] = 0$. Since $u$ is fixed, $\mathbb{E}[\langle u, q_t \rangle] = \langle u, \mathbb{E}[q_t] \rangle = 0$.

Now consider $\mathbb{E}[\langle u_t, q_t \rangle]$, notice that given $\eta_1, \ldots, \eta_{t-1}$ then $u_t$ is a constant. We have:

$$\begin{aligned}
\mathbb{E}[\langle u_t, q_t \rangle] &= \mathbb{E}[\mathbb{E}[\langle u_t, q_t \rangle | \eta_1, \eta_2, \ldots, \eta_{t-1}]] \\
&= \mathbb{E}[\langle u_t, \mathbb{E}[q_t | \eta_1, \eta_2, \ldots, \eta_{t-1}] \rangle] \\
&= \mathbb{E}[\langle u_t, 0 \rangle] = 0.
\end{aligned}$$

$\square$

Now we are going to bound the term $\sum_{t=1}^{T-1} \|\mathbf{A}(\eta_{t+1} - \eta_t)\|^2$.

**Lemma B.3.**

$$\sum_{t=1}^{T-1} \|\mathbf{A}(\eta_{t+1} - \eta_t)\|^2 \leq 6 + 3 \sum_{t=1}^{T-1} \|v_{t+1} - v_t\|^2, \tag{4}$$

$$\sum_{t=1}^{T-1} \|\mathbf{A}^\mathsf{T}(\zeta_{t+1} - \zeta_t)\|^2 \leq 6 + 3 \sum_{t=1}^{T-1} \|u_{t+1} - u_t\|^2. \tag{5}$$

*Proof.* Recall that by rescaling we have $\|\mathbf{A}\| \leq 1$. Hence,

$$\sum_{t=1}^{T-1} \|\mathbf{A}(\eta_{t+1} - \eta_t)\|^2 \leq \sum_{t=1}^{T-1} \|\eta_{t+1} - \eta_t\|^2.$$

Write $\eta_{t+1} - \eta_t = (\eta_{t+1} - v_{t+1}) + (v_{t+1} - v_t) + (v_t - \eta_t)$. Using the triangle inequality of the $l_1$ norm and the Cauchy inequality $(a + b + c)^2 \leq 3(a^2 + b^2 + c^2)$, we get

$$\sum_{t=1}^{T-1} \|\eta_{t+1} - \eta_t\|^2 \leq 6 \sum_{t=1}^{T} \|\eta_t - v_t\|^2 + 3 \sum_{t=1}^{T-1} \|v_{t+1} - v_t\|^2. \tag{6}$$

Similarly, we have:

$$\sum_{t=1}^{T-1} \|\zeta_{t+1} - \zeta_t\|^2 \leq 6 \sum_{t=1}^{T} \|\zeta_t - u_t\|^2 + 3 \sum_{t=1}^{T-1} \|u_{t+1} - u_t\|^2. \tag{7}$$

Observing that in our algorithm we collect $T$ independent and identically distributed samples and take their average, we have:

$$\mathbb{E}\left[\sum_{t=1}^{T} \|\zeta_t - u_t\|^2\right] \leq 1,$$

$$\mathbb{E}\left[\sum_{t=1}^{T} \|\eta_t - v_t\|^2\right] \leq 1.$$

Combining the result above, we just get the desired equation. $\square$

We also need the following lemma to guarantee that the sum of the regret is always non-negative.

**Lemma B.4.** *The sum of the regrets of two players in Algorithm 1 is always non-negative. In other words:*

$$\max_{u\in\Delta_m}\max_{v\in\Delta_n}\left(\sum_{t=1}^{T}\langle u_t - u, \mathbf{A}v_t\rangle + \sum_{t=1}^{T}\langle v_t - v, -\mathbf{A}^\mathsf{T}u_t\rangle\right) \geq 0.$$

*Proof.*

$$\max_{u\in\Delta_m}\max_{v\in\Delta_n}\left(\sum_{t=1}^{T}\langle u_t - u, \mathbf{A}v_t\rangle + \sum_{t=1}^{T}\langle v_t - v, -\mathbf{A}^\mathsf{T}u_t\rangle\right)$$

$$= \max_{u\in\Delta_m}\max_{v\in\Delta_n}\left(\sum_{t=1}^{T}\langle -u, \mathbf{A}v_t\rangle + \sum_{t=1}^{T}\langle v, \mathbf{A}^\mathsf{T}u_t\rangle\right)$$

$$= \max_{v\in\Delta_n}\sum_{t=1}^{T}\langle v, \mathbf{A}^\mathsf{T}u_t\rangle - \min_{u\in\Delta_m}\sum_{t=1}^{T}\langle u, \mathbf{A}v_t\rangle \geq 0$$

The last step is because

$$\max_{v\in\Delta_n}\sum_{t=1}^{T}\langle v, \mathbf{A}^\mathsf{T}u_t\rangle \geq \left\langle \mathbf{A}\sum_{t=1}^{T} v_t/T, \sum_{t=1}^{T} u_t \right\rangle,$$

and

$$\min_{u\in\Delta_m}\sum_{t=1}^{T}\langle u, \mathbf{A}v_t\rangle \leq \left\langle \mathbf{A}\sum_{t=1}^{T} v_t, \sum_{t=1}^{T} u_t/T \right\rangle.$$

$\square$

Combining the result above, we finally have the following theorem.

**Theorem B.5.** *Suppose that in our Algorithm 1, we choose the episode $T = \widetilde{\Theta}(1/\varepsilon)$, and choose a constant learning rate $\lambda$ that satisfies $\lambda < \sqrt{3}/6$. Then with probability at least $2/3$ the total regret of the algorithm is $\widetilde{O}(1)$. To be more clear, we have:*

$$T\left(\max_{v\in\Delta_n}\langle v, \mathbf{A}^\mathsf{T}\hat{u}\rangle - \min_{u\in\Delta_m}\langle u, \mathbf{A}\hat{v}\rangle\right) \leq 36\lambda + \frac{3\log(mn)}{\lambda},$$

*and so our algorithm returns an $\varepsilon$-approximate Nash equilibrium.*

*Proof.* Adding the inequalities (2) and (3) together, we get

$$\sum_{t=1}^{T}\langle u_t - u, \mathbf{A}\eta_t\rangle + \sum_{t=1}^{T}\langle v_t - v, -\mathbf{A}^\mathsf{T}\zeta_t\rangle \leq \frac{\log m}{\lambda} + \frac{\log n}{\lambda}$$

$$+ \lambda\sum_{t=1}^{T-1}\|\mathbf{A}(\eta_{t+1}-\eta_t)\|^2 - \frac{1}{4\lambda}\sum_{t=1}^{T-1}\|v_{t+1}-v_t\|^2 \tag{8}$$

$$+ \lambda\sum_{t=1}^{T-1}\|\mathbf{A}^\mathsf{T}(\zeta_{t+1}-\zeta_t)\|^2 - \frac{1}{4\lambda}\sum_{t=1}^{T-1}\|u_{t+1}-u_t\|^2.$$

Taking expectation, and using the inequalities (6) we have

$$\mathbb{E}\left[\lambda\sum_{t=1}^{T-1}\|\mathbf{A}(\eta_{t+1}-\eta_t)\|^2 - \frac{1}{4\lambda}\sum_{t=1}^{T-1}\|v_{t+1}-v_t\|^2\right]$$

$$\leq \left(3\lambda - \frac{1}{4\lambda}\right)\mathbb{E}\left[\sum_{t=1}^{T-1}\|v_{t+1}-v_t\|^2\right] + 6\lambda\cdot\mathbb{E}\left[\sum_{t=1}^{T}\|\eta_t - v_t\|^2\right]$$

$$\leq 6\lambda.$$

Similarly we can prove

$$\mathbb{E}\left[\lambda \sum_{t=1}^{T-1} \|\mathbf{A}^\mathsf{T}(\zeta_{t+1} - \zeta_t)\|^2 - \frac{1}{4\lambda} \sum_{t=1}^{T-1} \|u_{t+1} - u_t\|^2\right] \le 6\lambda.$$

So, taking expectations of Equation (8), and using the above inequalities and the Lemma B.2, we get

$$\mathbb{E}\left[\max_{u \in \Delta_m} \sum_{t=1}^{T} \langle u_t - u, \mathbf{A}v_t\rangle + \max_{v \in \Delta_n} \sum_{t=1}^{T} \langle v_t - v, -\mathbf{A}^\mathsf{T}u_t\rangle\right] \le 12\lambda + \frac{\log(mn)}{\lambda}. \qquad (9)$$

Using the fact that

$$\mathbb{E}[\hat{u}] \cdot T = \sum_{t=1}^{T} \mathbb{E}[u_t],$$

$$\mathbb{E}[\hat{v}] \cdot T = \sum_{t=1}^{T} \mathbb{E}[v_t],$$

we have

$$\mathbb{E}\left[\max_{v \in \Delta_n} \langle v, \mathbf{A}^\mathsf{T}\hat{u}\rangle - \min_{u \in \Delta_m} \langle u, \mathbf{A}\hat{v}\rangle\right] \cdot T \le 12\lambda + \frac{\log(mn)}{\lambda}. \qquad (10)$$

By Lemma B.4, we know that the regret is always non-negative. So applying Markov's inequality, we know with probability at least $2/3$, the following inequality holds:

$$\max_{v \in \Delta_n} \langle v, \mathbf{A}^\mathsf{T}\hat{u}\rangle - \min_{u \in \Delta_m} \langle u, \mathbf{A}\hat{v}\rangle \le \frac{1}{T}\left(36\lambda + \frac{3\log(mn)}{\lambda}\right).$$

$\square$

## B.2 Samplers with Errors

**Theorem B.6** (Restatement of Theorem 3.2). *Suppose that in our Algorithm 1, we choose the episode $T = \widetilde{O}(1/\varepsilon)$, and choose a constant learning rate $\lambda$ that satisfies $0 < \lambda < \sqrt{3}/6$. The quantum implementation of the oracle in the algorithm will return $T$ independent and identically distributed samples from a distribution that is $\epsilon_G$-close to the desired distribution in total variational distance in quantum time $T_G^Q$.*

*Then with probability at least $2/3$ the total regret of the algorithm is $\widetilde{O}(1 + \epsilon_G/\varepsilon)$ and the algorithm returns an $\widetilde{O}(\varepsilon + \epsilon_G)$-approximate Nash equilibrium in quantum time $\widetilde{O}(T_G^Q/\varepsilon)$.*

*Proof.* We will follow similar steps of proof for Theorem B.5. Since the sampling is not from the ideal distribution, we must bound the terms where $\eta_t$ and $\zeta_t$ take place.

Notice that in this case, we have

$$\|A(v_t - \mathbb{E}[\eta_t])\| \le \|v_t - \mathbb{E}[\eta_t]\| \le \epsilon_G.$$

So for the term $q_t$ in Lemma B.2 we now have the bound:

$$\mathbb{E}\left[\sum_{t=1}^{T} \langle u_t - u, A(v_t - \eta_t)\rangle\right]$$

$$= \mathbb{E}\left[\sum_{t=1}^{T} \langle u_t - u, A(v_t - \mathbb{E}[\eta_t])\rangle\right] + \mathbb{E}\left[\sum_{t=1}^{T} \langle u_t - u, A(\mathbb{E}[\eta_t] - \eta_t)\rangle\right]$$

$$= \mathbb{E}\left[\sum_{t=1}^{T} \langle u_t - u, A(v_t - \mathbb{E}[\eta_t])\rangle\right] \le 2T\epsilon_G,$$

where the last step is by Hölder's inequality.

Then for the other term, we have

$$\mathbb{E}\left[\sum_{t=1}^{T}\|\eta_t - v_t\|^2\right] \leq 2 \cdot \mathbb{E}\left[\sum_{t=1}^{T}\|\eta_t - \mathbb{E}[\eta_t]\|^2\right] + 2 \cdot \mathbb{E}\left[\sum_{t=1}^{T}\|v_t - \mathbb{E}[\eta_t]\|^2\right]$$
$$\leq 2 + 2T\epsilon_G^2.$$

So following the similar steps of proof for Theorem B.5, and using the above bounds, we can get

$$\mathbb{E}\left[\max_{u \in \Delta_m}\sum_{t=1}^{T}\langle u_t - u, \mathbf{A}v_t\rangle + \max_{v \in \Delta_n}\sum_{t=1}^{T}\langle v_t - v, -\mathbf{A}^\mathsf{T}u_t\rangle\right]$$
$$\leq 24\lambda + 24\lambda T\epsilon_G^2 + \frac{\log(mn)}{\lambda} + 4T\epsilon_G.$$

Again using linearity of expectation and Markov's inequality, we conclude that with probability at least $2/3$

$$T\left(\max_{v \in \Delta_n}\langle v, \mathbf{A}^\mathsf{T}\hat{u}\rangle - \min_{u \in \Delta_m}\langle u, \mathbf{A}\hat{v}\rangle\right) \leq 72\lambda + \frac{3\log(mn)}{\lambda} + 72T\lambda\epsilon_G^2 + 12T\epsilon_G.$$

$\square$

## C   Consistent Quantum Amplitude Estimation

**Theorem C.1** (Consistent phase estimation, [2, 47])**.** *Suppose $U$ is a unitary operator. For every positive reals $\epsilon, \delta$, there is a quantum algorithm (a unitary quantum circuit) $\mathcal{A}$ such that, on input $O\left(\log\left(\epsilon^{-1}\right)\right)$-bit random string $s$, it holds that*

- *For every eigenvector $|\psi_\theta\rangle$ of $U$ (where $U|\psi_\theta\rangle = \exp(i\theta)|\psi_\theta\rangle$), with probability $\geq 1 - \epsilon$:*
$$\langle\psi_\theta|\langle f(s,\theta)|\mathcal{A}|\psi_\theta\rangle|0\rangle \geq 1 - \epsilon;$$

- *$f(s,\theta)$ is a function of $s$ and $\theta$ such that $|f(s,\theta) - \theta| < \delta$,*

*with time complexity $\widetilde{O}\left(\delta^{-1}\right) \cdot \mathrm{poly}\left(\epsilon^{-1}\right)$.*

**Theorem C.2** (Consistent quantum amplitude estimation)**.** *Suppose $U$ is a unitary operator such that*

$$U|0\rangle_A|0\rangle_B = \sqrt{p}|0\rangle_A|\phi_0\rangle_B + \sqrt{1-p}|1\rangle_A|\phi_1\rangle_B.$$

*where $p \in [0,1]$ and $|\phi_0\rangle$ and $|\phi_1\rangle$ are normalized pure quantum states. Then for every positive reals $\epsilon, \delta$, there is a quantum algorithm that, on input $O\left(\log\left(\epsilon^{-1}\right)\right)$-bit random string $s$, outputs $f(s,p) \in [0,1]$ such that*
$$\mathbf{Pr}\left[|f(s,p) - p| \leq \delta\right] \geq 1 - \epsilon,$$
*with time complexity $\widetilde{O}\left(\delta^{-1}\right) \cdot \mathrm{poly}\left(\epsilon^{-1}\right)$.*

*Proof.* Suppose $U$ is a unitary operator such that

$$U|0\rangle_A|0\rangle_B = \sqrt{p}|0\rangle_A|\phi_0\rangle_B + \sqrt{1-p}|1\rangle_A|\phi_1\rangle_B.$$

Let

$$Q = -U(I - 2|0\rangle_A\langle 0| \otimes |0\rangle_B\langle 0|)U^\dagger(I - 2|0\rangle_A\langle 0| \otimes I_B).$$

Similar to the analysis in Brassard et al. [9], we have

$$U|0\rangle_A|0\rangle_B = \frac{-i}{\sqrt{2}}\left(\exp\left(i\theta_p\right)|\psi_+\rangle_{AB} - \exp\left(-i\theta_p\right)|\psi_-\rangle_{AB}\right),$$

where $\sin^2\left(\theta_p\right) = p$ ($0 \leq \theta_p < \pi/2$), and

$$|\psi_\pm\rangle_{AB} = \frac{1}{\sqrt{2}}\left(|0\rangle_A|\phi_0\rangle_B \pm i|1\rangle_A|\phi_1\rangle_B\right).$$

Note that $|\psi_\pm\rangle_{AB}$ are eigenvectors of $Q$, i.e., $Q|\psi_\pm\rangle_{AB} = \exp(\pm i2\theta_p)|\psi_\pm\rangle_{AB}$.

Now applying the algorithm $\mathcal{A}$ of consistent phase estimation of $Q$ by Theorem C.1 on state $U|0\rangle_A|0\rangle_B \otimes |0\rangle_C$ (with an $O(\log(\epsilon^{-1}))$-bit random string $s$), we obtain

$$\mathcal{A}(U|0\rangle_A|0\rangle_B \otimes |0\rangle_C) = \frac{-i}{\sqrt{2}}\left(\exp(i\theta_p)\mathcal{A}(|\psi_+\rangle_{AB}|0\rangle_C) - \exp(-i\theta_p)\mathcal{A}(|\psi_-\rangle_{AB}|0\rangle_C)\right).$$

Since each of $|\psi_\pm\rangle_{AB}$ is an eigenvector of $Q$, it holds that, with probability $\geq 1 - \epsilon$,

$$\langle\psi_\pm|_{AB}\langle f(s, \pm 2\theta_p)|_C\mathcal{A}(|\psi_\pm\rangle_{AB}|0\rangle_C) \geq 1 - \epsilon.$$

which implies that $\mathcal{A}(U|0\rangle_A|0\rangle_B \otimes |0\rangle_C)$ is $O(\sqrt{\epsilon})$-close to

$$\frac{-i}{\sqrt{2}}\left(\exp(i\theta_p)|\psi_+\rangle_{AB}|f(s, 2\theta_p)\rangle_C - \exp(-i\theta_p)|\psi_-\rangle_{AB}|f(s, -2\theta_p)\rangle_C\right)$$

in trace distance, where $\left|f(s, \pm 2\theta_p) \mp 2\theta_p\right| < \delta$. Measuring register $C$, we denote the outcome as $\gamma$, which will be either $f(s, 2\theta_p)$ or $f(s, -2\theta_p)$. Finally, output $\sin^2(\gamma/2)$ as the estimate of $p$ (which is consistent). Since $\sin^2(\cdot)$ is even and 2-Lipschitz, the additive error is bounded by

$$\left|\sin^2\left(\frac{\gamma}{2}\right) - p\right| \leq 2\left|\left|\frac{\gamma}{2}\right| - |\theta_p|\right| < \delta.$$

Note that $\mathcal{A}$ makes $\widetilde{O}(\delta^{-1}) \cdot \mathrm{poly}(\epsilon^{-1})$ queries to $Q$, thus our consistent amplitude estimation has quantum time complexity $\widetilde{O}(\delta^{-1}) \cdot \mathrm{poly}(\epsilon^{-1})$. $\qquad\square$

**Theorem C.3** (Error-Reduced Consistent quantum amplitude estimation). *Suppose $U$ is a unitary operator such that*
$$U|0\rangle_A|0\rangle_B = \sqrt{p}|0\rangle_A|\phi_0\rangle_B + \sqrt{1-p}|1\rangle_A|\phi_1\rangle_B.$$

*where $p \in [0, 1]$ and $|\phi_0\rangle$ and $|\phi_1\rangle$ are normalized pure quantum states. Then for every positive integers $r$ and positive real $\delta$, there is a quantum algorithm that, on input $O(r)$-bit random string $s$, outputs $f^*(s, p) \in [0, 1]$ such that*

$$\mathbf{Pr}\left[|f^*(s, p) - p| \leq \delta\right] \geq 1 - O(\exp(-r)),$$

*with time complexity $\widetilde{O}(\delta^{-1}) \cdot \mathrm{poly}(r)$.*

*Proof.* Consider that we divide the input random string $s$ into $r$ strings $s_1, s_2, \ldots, s_r$ of length $O(1)$. For each $i \in [r]$, we use Theorem C.2 with input string $s_i$ and parameter $\epsilon = 1/10$. So we get, for each $i \in [r]$,

$$\mathbf{Pr}\left[|f(s_i, p) - p| \leq \delta\right] \geq \frac{9}{10}.$$

Now we set $f^*(s, p)$ to be the median of the estimations $f(s_i, p)$ for $i \in [r]$. We claim it satisfies the desired property. To show that, we define random variables $X_i$ for $i \in [r]$ as follows:

$$X_i = \begin{cases} 1, & \text{if } |f(s_i, p) - p| \leq \delta, \\ 0, & \text{otherwise.} \end{cases}$$

Noticing $\mathbb{E}\left[\sum_{i=1}^r X_i\right] \geq 9r/10$, and by Chernoff bound, we have:

$$\mathbf{Pr}\left[\sum_{i=1}^r X_i < \frac{r}{2}\right] \leq \exp\left(-\frac{8r}{45}\right).$$

Thus with probability at least $1 - \exp(-8r/45)$, we know that at least half of the estimations fall into the interval $[p - \delta, p + \delta]$, and then $f^*(s, p)$ returns a correct answer. $\qquad\square$

# D Details and Proofs of Fast Quantum Multi-Gibbs Sampling

We present the detailed version of the fast quantum multi-Gibss sampling. Here, we use the shorthand $\mathcal{O}_p^{\text{Gibbs}} = \mathcal{O}_p^{\text{Gibbs}}(1,0)$, and it also means the distribution of the sample.

We first define the notion of amplitude-encoding (a unitary operator that encodes a vector in its amplitudes).

**Definition D.1** (Amplitude-encoding). A unitary operator $V$ is said to be a $\beta$-amplitude-encoding of a vector $u \in \mathbb{R}^n$ with non-negative entries, if

$$\langle 0|_C V |0\rangle_C |i\rangle_A |0\rangle_B = \sqrt{\frac{u_i}{\beta}} |i\rangle_A |\psi_i\rangle_B$$

for all $i \in [n]$.

Then, as shown in Algorithm 4, we can construct a quantum multi-Gibbs sampler for a vector $u$ if an amplitude-encoding of the vector $u$ is given. To complete the proof of Theorem 4.2, we only have to construct an amplitude-encoding of $\mathbf{A}z$ (see Appendix D.2 for details).

---

**Algorithm 4** Quantum Multi-Gibbs Sampling implementing $\mathcal{O}_u^{\text{Gibbs}}(k, \epsilon_{\text{G}})$

---

**Input:** Sample count $k$, a $\beta$-amplitude-encoding $V$ of vector $u \in \mathbb{R}^n$, polynomial $P_{2\beta} \in \mathbb{R}[x]$ that satisfies Lemma 4.1 with parameter $\epsilon_P = k\epsilon_{\text{G}}^2/300n$.

**Output:** $k$ independent samples $i_1, i_2, \ldots, i_k$.

1: Obtain $\mathcal{O}_{\tilde{u}} \colon |i\rangle|0\rangle \mapsto |i\rangle|\tilde{u}_i\rangle$ using $\widetilde{O}(\beta)$ queries to $V$, where $u_i \leq \tilde{u}_i \leq u_i + 1$, by consistent quantum amplitude estimation (Theorem C.3).
2: Find the $k$ largest $\tilde{u}_i$'s by quantum $k$-maximum finding (Theorem D.3) and let $S$ be the set of their indexes. This can be done with $\widetilde{O}(\sqrt{nk})$ queries to $\mathcal{O}_{\tilde{u}}$.
3: Compute $\tilde{u}^* = \min_{i \in S} \tilde{u}_i$, and $W = (n-k)\exp(\tilde{u}^*) + \sum_{i \in S} \exp(\tilde{u}_i)$.
4: **for** $\ell = 1, \ldots, k$ **do**
5:  Prepare the quantum state

$$|u_{\text{guess}}\rangle = \sum_{i \in S} \sqrt{\frac{\exp(\tilde{u}_i)}{W}} |i\rangle + \sum_{i \notin S} \sqrt{\frac{\exp(\tilde{u}^*)}{W}} |i\rangle.$$

6:  Obtain $U_u = (V_{CAB}^\dagger \otimes I_D)(V_{DAB} \otimes I_C)$ being a block-encoding of $\text{diag}(u)/\beta$. Similarly, obtain $U_{\tilde{u}}^{\max}$ being a block-encoding of $\text{diag}(\max\{\tilde{u}, \tilde{u}^*\})/2\beta$.
7:  Obtain $U^-$ being a block-encoding of $\text{diag}(u - \max\{\tilde{u}, \tilde{u}^*\})/4\beta$ by the LCU (Linear-Combination-of-Unitaries) technique (Theorem D.6), using $O(1)$ queries to $U_u$ and $U_{\tilde{u}}^{\max}$.
8:  Obtain $U^{\exp}$ being a block-encoding of $P_{2\beta}(\text{diag}(u - \max\{\tilde{u}, \tilde{u}^*\})/4\beta)$ by the QSVT technique (Theorem D.7), using $O(\beta \log(\epsilon_P^{-1}))$ queries to $U^-$.
9:  Post-select $|\tilde{u}_{\text{post}}\rangle = \langle 0|^{\otimes a} U^{\exp} |u_{\text{guess}}\rangle|0\rangle^{\otimes a}$ by quantum amplitude amplification (Theorem D.8), and obtain $|\tilde{u}_{\text{Gibbs}}\rangle = |\tilde{u}_{\text{post}}\rangle/\||\tilde{u}_{\text{post}}\rangle\|$. (Suppose $U^{\exp}$ has $a$ ancilla qubits.)
10:  Measure $|\tilde{u}_{\text{Gibbs}}\rangle$ in the computational basis and let $i_\ell \in [n]$ be the outcome.
11: **end for**
12: **Return** $i_1, i_2, \ldots, i_k$.

---

## D.1 Useful Theorems

**Theorem D.2** (Quantum state preparation, [23, 32]). *There is a data structure implemented on QRAM maintaining an array $a_1, a_2, \ldots, a_\ell$ of positive numbers that supports the following operations.*

- *Initialization: For any value $c$, set $a_i \leftarrow c$ for all $i \in [\ell]$.*

- *Assignment: For any index $i$ and value $c$, set $a_i \leftarrow c$.*

- *State Preparation: Prepare a quantum state*

$$|a\rangle = \sum_{i \in [\ell]} \sqrt{\frac{a_i}{\|a\|_1}} |i\rangle.$$

*Each operation costs* $\mathrm{polylog}(\ell)$ *time.*

**Theorem D.3** (Quantum $k$-maximum finding, Theorem 6 of Dürr et al. [19]). *Given $k \in [n]$ and quantum oracle $\mathcal{O}_u$ for an array $u_1, u_2, \ldots, u_n$, i.e., for every $i \in [n]$,*

$$\mathcal{O}_u |i\rangle |0\rangle = |i\rangle |u_i\rangle,$$

*there is a quantum algorithm that, with probability $\geq 0.99$, finds a set $S \subseteq [n]$ of cardinality $|S| = k$ such that $u_i \geq u_j$ for every $i \in S$ and $j \notin S$, using $O\left(\sqrt{nk}\right)$ queries to $\mathcal{O}_u$.*

We now recall the definition of block-encoding, a crucial concept in quantum singular value transformation [20], which is used in line 9 to 12 in Algorithm 4.

**Definition D.4** (Block-encoding). *Suppose $A$ is a linear operator on $b$ qubits, $\alpha, \epsilon \geq 0$ and $a$ is a positive integer. A $(b + a)$-qubit unitary operator $U$ is said to be an $(\alpha, \epsilon)$-block-encoding of $A$, if*

$$\left\| \alpha \langle 0 |^{\otimes a} U | 0 \rangle^{\otimes a} - A \right\|_{\mathrm{op}} \leq \epsilon.$$

**Definition D.5** (State Preparation Pair, Definition 28 of Gilyén et al. [20]). *Let $y \in \mathbb{R}^n$ be a vector, specially in this context the number of coordinates starts from 0. Suppose $\|y\|_1 \leq \beta$. Let $\epsilon$ be a positive real. We call a pair of unitaries $(P_L, P_R)$ acting on $b$ qubits a $(\beta, \epsilon)$-state-preparation pair for $y$ if*

$$P_L |0\rangle^{\otimes b} = \sum_{j=0}^{2^b - 1} c_j |j\rangle,$$

$$P_R |0\rangle^{\otimes b} = \sum_{j=0}^{2^b - 1} d_j |j\rangle,$$

*such that:*

$$\sum_{j=0}^{m-1} \left| \beta c_j^* d_j - y_j \right| \leq \epsilon$$

*and for $j \in [2^b]$, $j \geq m$, we require $c_j^* d_j = 0$.*

We now state a theorem about linear combination of unitary operators, introduced by Berry et al. [5] and Childs and Wiebe [15]. The following form is from Gilyén et al. [20]. Again we restrict ourselves to the case of real linear combinations.

**Theorem D.6** (Linear Combination of Unitaries, Lemma 29 of Gilyén et al. [20]). *Let $\epsilon$ be a positive real number and $y \in \mathbb{R}^n$ be a vector as in Definition D.5 with $(\beta, \epsilon_1)$ state preparation pair $(P_L, P_R)$. Let $\{A_j\}_{j=0}^{m-1}$ be a set of linear operators on $s$ qubits, and forall $j$, we have $U_j$ as an $(\alpha, \epsilon_2)$-block-encoding of $A_j$ acting on $a + s$ qubits. Let*

$$W = \left( \sum_{j=0}^{m-1} |j\rangle \langle j| \otimes U_j \right) + \left( I - \sum_{j=0}^{m-1} |j\rangle \langle j| \right) \otimes I_{a+s},$$

*Then we can implement a $(\alpha\beta, \alpha\epsilon_1 + \alpha\beta\epsilon_2)$-block-encoding of $A = \sum_{j=0}^{m-1} y_j A_j$, with one query from $P_L^\dagger$, $P_R$, and $W$.*

**Theorem D.7** (Eigenvalue transformation, Theorem 31 of Gilyén et al. [20]). *Suppose $U$ is an $(\alpha, \epsilon)$-block-encoding of an Hermitian operator $A$. For every $\delta > 0$ and real polynomial $P \in \mathbb{R}[x]$ of degree $d$ such that $|P(x)| \leq 1/2$ for all $x \in [-1, 1]$, there is an efficiently computable quantum circuit $\tilde{U}$, which is a $\left( 1, 4d\sqrt{\epsilon/\alpha} + \delta \right)$-block-encoding of $P(A/\alpha)$, using $O(d)$ queries to $U$.*

Finally, for quantum amplitude amplification without knowing the exact value of the amplitude, we need the following theorem:

**Theorem D.8** (Quantum amplitude amplification, Theorem 3 of Brassard et al. [9])**.** *Suppose $U$ is a unitary operator such that*

$$U|0\rangle_A|0\rangle_B = \sqrt{p}|0\rangle_A|\phi_0\rangle_B + \sqrt{1-p}|1\rangle_A|\phi_1\rangle_B.$$

*where $p \in [0,1]$ is unknown and $|\phi_0\rangle$ and $|\phi_1\rangle$ are normalized pure quantum states. There is a quantum algorithm that outputs $|0\rangle_A|\phi_0\rangle_B$ with probability $\geq 0.99$, using $O(1/\sqrt{p})$ queries to $U$.*

### D.2  Main Proof

We generalize Theorem 4.2 as follows.

**Theorem D.9.** *Algorithm 4 will produce $k$ independent and identical distributed samples from a distribution that is $\epsilon_G$-close to $\mathcal{O}_u^{\text{Gibbs}}$ in total variation distance, in quantum time $\widetilde{O}\left(\beta\sqrt{nk}\right)$.*

It is immediate to show Theorem 4.2 from Theorem D.9 by constructing a $\beta$-amplitude-encoding $V$ of $\mathbf{A}z$. To see this, let $u = \mathbf{A}z$, then $u_i = (\mathbf{A}z)_i \in [0, \beta]$. By Theorem D.2, we can implement a unitary operator $U_z^{\text{QRAM}}$ such that

$$U_z^{\text{QRAM}}: |0\rangle_C|0\rangle_B \mapsto |0\rangle_C \sum_{j\in[n]} \sqrt{\frac{z_j}{\beta}}|j\rangle_B + |1\rangle_C|\phi\rangle_B.$$

Using two queries to $\mathcal{O}_\mathbf{A}$, we can construct a unitary operator $\mathcal{O}'_\mathbf{A}$ such that

$$\mathcal{O}'_\mathbf{A}: |0\rangle_E|i\rangle_A|j\rangle_B \mapsto \left(\sqrt{A_{i,j}}|0\rangle_E + \sqrt{1-A_{i,j}}|1\rangle_E\right)|i\rangle_A|j\rangle_B.$$

Let

$$V = \left(|0\rangle_C\langle 0| \otimes \mathcal{O}'_\mathbf{A} + |1\rangle_C\langle 1| \otimes I_{EAB}\right)\left(U_z^{\text{QRAM}} \otimes I_{EA}\right). \tag{11}$$

It can be verified (see Proposition D.10) that

$$\langle 0|_C\langle 0|_E V|0\rangle_C|0\rangle_E|i\rangle_A|0\rangle_B = \sum_{j\in[n]} \sqrt{\frac{A_{i,j}z_j}{\beta}}|i\rangle_A|j\rangle_B,$$

and thus $\langle 0|_C\langle 0|_E V|0\rangle_C|0\rangle_E|i\rangle_A|0\rangle_B = \sqrt{u_i/\beta}|i\rangle_A|\psi_i\rangle_B$ for some $|\psi_i\rangle$. Therefore, $V$ is a $\beta$-amplitude-encoding of $\mathbf{A}z$.

Now, we will show Theorem D.9 in the following.

*Proof of Theorem D.9.* Now we start to describe our algorithm. By our consistent quantum amplitude estimation (Theorem C.3), we choose an $O(r)$-bit random string $s$, then we can obtain a quantum algorithm $\mathcal{O}_{\hat{u}}$ such that, with probability $1 - O(\exp(-r))$, for every $i \in [n]$, it computes $f^*(s, u_i/\beta)$ with $\widetilde{O}(\delta^{-1}) \cdot \text{poly}(r)$ queries to $V$, where $f^*(s, p)$ is a function that only depends on $s$ and $p$, and it holds that

$$|f^*(s,p) - p| \leq \delta$$

for every $p \in [-1, 1]$. Here, $r, \delta$ are parameters to be determined. Note that

$$\frac{u_i}{\beta} = \|\langle 0|_C V|0\rangle_C|i\rangle_A|0\rangle_B\|^2,$$

so when applying consistent quantum amplitude estimation, we just use a controlled-XOR gate conditioned on the index and with $A$ the target system, before every query to $V$.

By quantum $k$-maximum finding algorithm (Theorem D.3), we can find a set $S \subseteq [n]$ with $|S| = k$ such that $f^*(s, u_i/\beta) \geq f^*(s, u_j/\beta)$ for every $i \in S$ and $j \notin S$ with probability $0.99 - O\left(\sqrt{nk}\exp(-r)\right)$, using $O\left(\sqrt{nk}\right)$ queries to $\mathcal{O}_{\hat{u}}$. To obtain a constant probability, it is sufficient to choose $r = \Theta(\log(n))$.

For each $i \in S$, again applying our consistent quantum amplitude estimation (Theorem C.3), we can obtain the value of $f^*(s, u_i/\beta)$ with probability $1 - O(\exp(-r))$, using $\widetilde{O}(\delta^{-1}) \cdot \text{poly}(r)$ queries to $V$; then we set

$$\hat{u}_i = \beta f^*\left(s, \frac{u_i}{\beta}\right)$$

for all $i \in S$, with success probability $1 - O(k \exp(-r))$ and using $\widetilde{O}(k\delta^{-1}) \cdot \text{poly}(r)$ queries to $V$ in total. It can be seen that $|\hat{u}_i - u_i| \leq \beta\delta$ for every $i \in S$.

Let $\tilde{u}_i = \hat{u}_i + \beta\delta$, and then we store $\tilde{u}_i$ for all $i \in S$ in the data structure as in Theorem D.2 (which costs $O(k)$ QRAM operations). Then, we calculate

$$W = (n-k)\exp(\tilde{u}^*) + \sum_{i \in S} \exp(\tilde{u}_i)$$

by classical computation in $\widetilde{O}(k)$ time, where

$$\tilde{u}^* = \min_{i \in S} \tilde{u}_i.$$

By Theorem D.2, we can prepare the quantum state

$$|u_{\text{guess}}\rangle = \sum_{i \in S} \sqrt{\frac{\exp(\tilde{u}_i)}{W}} |i\rangle + \sum_{i \notin S} \sqrt{\frac{\exp(\tilde{u}^*)}{W}} |i\rangle$$

in $\widetilde{O}(1)$ time.

Now we introduce another system $D$, and then let

$$U_u = (V_{CAB}^\dagger \otimes I_D)(V_{DAB} \otimes I_C).$$

It can be shown (see Proposition D.11) that $U_u$ is a $(1,0)$-block-encoding of $\text{diag}(u)/\beta$. By QRAM access to $\tilde{u}_i$, we can implement a unitary operator

$$V_{\tilde{u}} \colon |i\rangle_A |0\rangle_B \mapsto |i\rangle_A \left( \sqrt{\frac{\max\{\tilde{u}_i, \tilde{u}^*\}}{2\beta}} |0\rangle_B + \sqrt{1 - \frac{\max\{\tilde{u}_i, \tilde{u}^*\}}{2\beta}} |1\rangle_B \right)$$

in $\widetilde{O}(1)$ time by noting that $\max\{\tilde{u}_i, \tilde{u}^*\} = \tilde{u}_i$ if $i \in S$ and $\tilde{u}^*$ otherwise. We introduce one-qubit system $C$, and let

$$U_{\tilde{u}}^{\max} = \left( V_{\tilde{u}}^\dagger \otimes I_C \right)(\text{SWAP}_{BC} \otimes I_A)(V_{\tilde{u}} \otimes I_C).$$

It can be shown that $U_{\tilde{u}}^{\max}$ is a $(1,0)$-block-encoding of $\text{diag}(\max\{\tilde{u}, \tilde{u}^*\})/2\beta$. Applying the LCU technique (Theorem D.6), we can obtain a unitary operator $U^-$ that is a $(1,0)$-block-encoding of $\text{diag}(u - \max\{\tilde{u}, \tilde{u}^*\})/4\beta$, using $O(1)$ queries to $U_u$ and $U_{\tilde{u}}^{\max}$. By the QSVT technique (Theorem D.7 and Lemma 4.1), we can construct a unitary operator $U^{\exp}$ that is a $(1,0)$-block-encoding of $P_{2\beta}(\text{diag}(u - \max\{\tilde{u}, \tilde{u}^*\})/4\beta)$, using $O(\beta \log(\epsilon_P^{-1}))$ queries to $U^-$, where

$$\left| P_{2\beta}(x) - \frac{1}{4}\exp(2\beta x) \right| \leq \epsilon_P$$

for every $x \in [-1, 0]$ and $\epsilon_P \in (0, 1/2)$ is to be determined. Suppose $U^{\exp}$ has an $a$-qubit ancilla system, and let $|\tilde{u}_{\text{post}}\rangle = \langle 0|^{\otimes a} U^{\exp} |u_{\text{guess}}\rangle |0\rangle^{\otimes a}$. Note that

$$|\tilde{u}_{\text{post}}\rangle = \sum_{i \in S} P_{2\beta}\left(\frac{u_i - \tilde{u}_i}{4\beta}\right) \sqrt{\frac{\exp(\tilde{u}_i)}{W}} |i\rangle + \sum_{i \notin S} P_{2\beta}\left(\frac{u_i - \tilde{u}^*}{4\beta}\right) \sqrt{\frac{\exp(\tilde{u}^*)}{W}} |i\rangle.$$

It can be shown (Proposition D.12) that $\| |\tilde{u}_{\text{post}}\rangle \|^2 \geq \Theta(k/n)$; thus by quantum amplitude amplification (Theorem D.8), we can obtain

$$|\tilde{u}_{\text{Gibbs}}\rangle = \frac{|\tilde{u}_{\text{post}}\rangle}{\| |\tilde{u}_{\text{post}}\rangle \|}$$

using $O(\sqrt{n/k})$ queries to $U^{\mathrm{exp}}$. By measuring $|\tilde{u}_{\mathrm{Gibbs}}\rangle$ on the computational basis, we return the outcome as a sample from the distribution $\tilde{u}_{\mathrm{Gibbs}}$; it can be shown (Proposition D.13) that the total variation distance between $\tilde{u}_{\mathrm{Gibbs}}$ and $\mathcal{O}_u^{\mathrm{Gibbs}}$ is bounded by

$$d_{\mathrm{TV}}\left(\tilde{u}_{\mathrm{Gibbs}}, \mathcal{O}_u^{\mathrm{Gibbs}}\right) \leq \sqrt{\frac{88n\epsilon_P}{k\exp(-2\beta\delta)}}.$$

By taking $\delta = 1/2\beta$ and $\epsilon_P = k\epsilon_{\mathrm{G}}^2/300n$, we can produce one sample from $\tilde{u}_{\mathrm{Gibbs}}$, using $\widetilde{O}(\beta\sqrt{n/k})$ queries to $U_u$ and $U_{\tilde{u}}^{\mathrm{max}}$, with $\widetilde{O}(\beta\sqrt{nk})$-time precomputation.

Finally, by applying $k$ times the above procedure (with the precomputation processed only once), we can produce $k$ independent and identically distributed samples from $\tilde{u}_{\mathrm{Gibbs}}$ that is $\epsilon_{\mathrm{Gibbs}}$-close to the Gibbs distribution $\mathcal{O}_u^{\mathrm{Gibbs}}$, with total time complexity

$$\widetilde{O}\left(\beta\sqrt{nk}\right) + k \cdot \widetilde{O}\left(\beta\sqrt{\frac{n}{k}}\right) = \widetilde{O}\left(\beta\sqrt{nk}\right).$$

$\square$

## D.3 Technical Lemmas

**Proposition D.10.** *Let $V$ defined by Equation* (11)*, we have*

$$\langle 0|_C\langle 0|_D V|0\rangle_C|0\rangle_D|i\rangle_A|0\rangle_B = \sum_{j\in[n]}\sqrt{\frac{A_{i,j}z_j}{\beta}}|i\rangle_A|j\rangle_B.$$

*Proof.*

$V|0\rangle_C|0\rangle_D|i\rangle_A|0\rangle_B$

$= \left(|0\rangle_C\langle 0| \otimes \mathcal{O}_{\mathbf{A}}' + |1\rangle_C\langle 1| \otimes I_{AB}\right)\left(|0\rangle_C|0\rangle_D|i\rangle_A\sum_{j\in[n]}\sqrt{\frac{z_j}{\beta}}|j\rangle_B + |1\rangle_C|0\rangle_D|i\rangle_A|\phi\rangle_B\right)$

$= |0\rangle_C\sum_{j\in[n]}\left(\sqrt{A_{i,j}}|0\rangle_D + \sqrt{1-A_{i,j}}|1\rangle_D\right)\sqrt{\frac{z_j}{\beta}}|i\rangle_A|j\rangle_B + |1\rangle_C|0\rangle_D|i\rangle_A|\phi\rangle_B$

$= |0\rangle_C|0\rangle_D\sum_{j\in[n]}\sqrt{\frac{A_{i,j}z_j}{\beta}}|i\rangle_A|j\rangle_B + |0\rangle_C|1\rangle_D\sum_{j\in[n]}\sqrt{\frac{(1-A_{i,j})z_j}{\beta}}|i\rangle_A|j\rangle_B + |1\rangle_C|0\rangle_D|i\rangle_A|\phi\rangle_B.$

$\square$

**Proposition D.11.** *In the proof of Theorem D.9, $U_u$ is a $(1,0)$-block-encoding of $\mathrm{diag}(u)/\beta$.*

*Proof.* To see this, for every $i,j \in [n]$,

$\langle 0|_C\langle 0|_D\langle j|_A\langle 0|_B U_u|0\rangle_C|0\rangle_D|i\rangle_A|0\rangle_B$

$= \langle 0|_C\langle 0|_D\langle j|_A\langle 0|_B(V_{CAB}^\dagger \otimes I_D)(V_{DAB} \otimes I_C)|0\rangle_C|0\rangle_D|i\rangle_A|0\rangle_B$

$= \left(\sqrt{u_j/\beta}\langle 0|_C\langle 0|_D\langle j|_A\langle\psi_j|_B + \langle 1|_C\langle 0|_D\langle g_j|_{AB}\right)\left(\sqrt{u_i/\beta}|0\rangle_C|0\rangle_D|i\rangle_A|\psi_i\rangle_B + |0\rangle_C|1\rangle_D|g_i\rangle_{AB}\right)$

$= \langle j|i\rangle_A\frac{u_i}{\beta}.$

$\square$

**Proposition D.12.** *In the proof of Theorem D.9, if $\delta = 1/2\beta$, $E = \sum_{j\in[n]}\exp(u_j)$, and $\epsilon_P = k\epsilon_{\mathrm{G}}^2/300n$, then*

$$\Theta\left(\frac{k}{n}\right) \leq \frac{E}{16W} - 2\epsilon_P \leq \||u_{\mathrm{post}}\rangle\|^2 \leq \frac{E}{16W} + 3\epsilon_P.$$

*Proof.* We first give an upper bound for $W$ in terms of $u_i$ and $\tilde{u}^*$. Notice that $\tilde{u}_i \leq u_i + 2\beta\delta$ for all $i \in S$, we have:

$$W = (n-k)\exp(\tilde{u}^*) + \sum_{i\in S}\exp(\tilde{u}_i) \leq \exp(2\beta\delta)\left((n-k)\exp(u^*) + \sum_{i\in S}\exp(u_i)\right).$$

Note that

$$\frac{(n-k)\exp(u^*) + \sum\limits_{i\in S}\exp(u_i)}{\sum\limits_{i\in[n]}\exp(u_i)} \leq \frac{n-k}{k} + 1 = \frac{n}{k},$$

then we have

$$\frac{E}{W} \geq \sum_{i\in[n]}\frac{\exp(u_i)}{\exp(2\beta\delta)((n-k)\exp(u^*) + \sum_{i\in S}\exp(u_i))} \geq \frac{k}{n}\exp(-2\beta\delta). \qquad (12)$$

With this, noting that $(a-b)^2 \geq a^2 - 2ab$ for any real $a$ and $b$, we have

$$\begin{aligned}
\||u_{\text{post}}\rangle\|^2 &= \sum_{i\in S}\left(P_{2\beta}\left(\frac{u_i-\tilde{u}_i}{4\beta}\right)\right)^2\frac{\exp(\tilde{u}_i)}{W} + \sum_{i\notin S}\left(P_{2\beta}\left(\frac{u_i-\tilde{u}^*}{4\beta}\right)\right)^2\frac{\exp(\tilde{u}^*)}{W} \\
&\geq \sum_{i\in S}\left(\left(\frac{1}{4}\exp\left(\frac{u_i-\tilde{u}_i}{2}\right)\right)^2 - 2\epsilon_P\right)\frac{\exp(\tilde{u}_i)}{W} \\
&\quad + \sum_{i\notin S}\left(\left(\frac{1}{4}\exp\left(\frac{u_i-\tilde{u}^*}{2}\right)\right)^2 - 2\epsilon_P\right)\frac{\exp(\tilde{u}^*)}{W} \\
&= \frac{1}{16}\left(\sum_{i\in S}\exp(u_i-\tilde{u}_i)\frac{\exp(\tilde{u}_i)}{W} + \sum_{i\notin S}\exp(u_i-\tilde{u}^*)\frac{\exp(\tilde{u}^*)}{W}\right) \\
&\quad - 2\epsilon_P\left(\sum_{i\in S}\frac{\exp(\tilde{u}_i)}{W} + \sum_{i\notin S}\frac{\exp(\tilde{u}^*)}{W}\right) \\
&\geq \frac{E}{16W} - 2\epsilon_P \\
&\geq \Theta\left(\frac{k}{n}\right).
\end{aligned}$$

On the other hand, a similar argument using the inequality $(a+b)^2 \leq a^2 + 3ab$ for positive real $a \geq b$ gives

$$\||u_{\text{post}}\rangle\|^2 \leq \frac{E}{16W} + 3\epsilon_P.$$

These yield the proof. $\qquad\square$

**Proposition D.13.** *In the proof of Theorem D.9, the total variation distance between the two distributions $\tilde{u}_{Gibbs}$ and $\mathcal{O}_u^{Gibbs}$ is bounded by*

$$d_{TV}\left(\tilde{u}_{Gibbs}, \mathcal{O}_u^{Gibbs}\right) \leq \sqrt{\frac{88n\epsilon_P}{k\exp(-2\beta\delta)}}.$$

*Proof.* Define $E = \sum\limits_{j\in[n]}\exp(u_j)$. Let

$$|u_{\text{Gibbs}}\rangle = \sum_{i\in[n]}\sqrt{\frac{\exp(u_i)}{E}}|i\rangle$$

be the intended quantum state with amplitudes the same as the Gibbs distribution $\mathcal{O}_u^{\text{Gibbs}}$. The inner product between $|\tilde{u}_{\text{post}}\rangle$ and $|u_{\text{Gibbs}}\rangle$ can be bounded by:

$$
\langle \tilde{u}_{\text{post}} | u_{\text{Gibbs}} \rangle = \sum_{i \in S} P_{2\beta} \left( \frac{u_i - \tilde{u}_i}{4\beta} \right) \sqrt{\frac{\exp(\tilde{u}_i)}{W}} \sqrt{\frac{\exp(u_i)}{E}}
$$

$$
+ \sum_{i \notin S} P_{2\beta} \left( \frac{u_i - \tilde{u}^*}{4\beta} \right) \sqrt{\frac{\exp(\tilde{u}^*)}{W}} \sqrt{\frac{\exp(u_i)}{E}}
$$

$$
\geq \sum_{i \in S} \left( \frac{1}{4} \exp\left( \frac{u_i - \tilde{u}_i}{2} \right) - \epsilon_P \right) \sqrt{\frac{\exp(\tilde{u}_i)}{W}} \sqrt{\frac{\exp(u_i)}{E}}
$$

$$
+ \sum_{i \notin S} \left( \frac{1}{4} \exp\left( \frac{u_i - \tilde{u}_i}{2} \right) - \epsilon_P \right) \sqrt{\frac{\exp(\tilde{u}^*)}{W}} \sqrt{\frac{\exp(u_i)}{E}}
$$

$$
\geq \frac{1}{4\sqrt{WE}} \left( \sum_{i \in [n]} \exp(u_i) \right) - \epsilon_P.
$$

The last step is by Cauchy's inequality. By Proposition D.12 and Equation (12), we have

$$
|\langle \tilde{u}_{\text{Gibbs}} | u_{\text{Gibbs}} \rangle|^2 = \frac{|\langle \tilde{u}_{\text{post}} | u_{\text{Gibbs}} \rangle|^2}{\| |\tilde{u}_{\text{post}}\rangle \|^2} \geq \frac{E}{16W \| |\tilde{u}_{\text{post}}\rangle \|^2} - \frac{\epsilon_P}{2 \| |\tilde{u}_{\text{post}}\rangle \|^2}
$$

$$
\geq \frac{E}{16W \left( \dfrac{E}{16W} + 3\epsilon_P \right)} - \frac{\epsilon_P}{2 \| |\tilde{u}_{\text{post}}\rangle \|^2}
$$

$$
\geq 1 - \frac{48\epsilon_P}{E/W} - \frac{8\epsilon_P}{E/W - 32\epsilon_P}
$$

$$
\geq 1 - \frac{48n\epsilon_P}{k\exp(-2\beta\delta)} - \frac{8n\epsilon_P}{k\exp(-2\beta\delta) - 32n\epsilon_P}
$$

$$
\geq 1 - \frac{88n\epsilon_P}{k\exp(-2\beta\delta)}.
$$

Finally, we have

$$
d_{\text{TV}}\left( \tilde{u}_{\text{Gibbs}}, \mathcal{O}_u^{\text{Gibbs}} \right) \leq \frac{1}{2} \text{tr}\left( \left| |\tilde{u}_{\text{Gibbs}}\rangle \langle \tilde{u}_{\text{Gibbs}}| - |u_{\text{Gibbs}}\rangle \langle u_{\text{Gibbs}}| \right| \right)
$$

$$
= \sqrt{1 - |\langle \tilde{u}_{\text{Gibbs}} | u_{\text{Gibbs}} \rangle|^2}
$$

$$
\leq \sqrt{\frac{88n\epsilon_P}{k\exp(-2\beta\delta)}},
$$

which is bounded by $\epsilon_G$ by the choice of $\epsilon_P$. $\qquad\square$

