# OpenReview forum: "Logarithmic-Regret Quantum Learning Algorithms for Zero-Sum Games"
_NeurIPS.cc/2023/Conference — NeurIPS 2023 poster_

### Official Review · Reviewer_8qKg · 2023-06-14

**Soundness:** 3 good
**Presentation:** 4 excellent
**Contribution:** 3 good
**Rating:** 7
**Confidence:** 3

**Summary:**

The paper presents an improved quantum online learning algorithm for approximating the Nash equilibrium of a zero-sum game. Notably, this is the first algorithm to achieve $\tilde{O}(1)$ regret with quantum speedup. The presented algorithm can then be applied to linear programming problems using the primal-dual approach. Additional contribution of the paper is a sampling method for Gibbs distribution for obtaining multiple samples with quantum speedups on both the number of samples and the number of possible outcomes.

**Strengths:**

- Multiple novel contributions on quantum algorithms: speedups for multi-Gibbs sampling and logarithmic regret Nash equilibrium approximation
- The paper is well-written and easy to follow

**Weaknesses:**

While the presented theoretical results are interesting, I consider the necessity of QRAM as a minor weakness since (to my knowledge) it is still a highly theoretical model whose physical realizability is unclear.

**Questions:**

The presented algorithm requires QRAM to work. How does this compare to the quantum algorithms of regret $\tilde{O}(\sqrt{T})$, i.e., do they too require QRAM?

Typos:
- Line 285 should probably say "Line 7"

**Limitations:**

I do not consider the work to have any serious limitations other than the ones that follow from the limited practicality of quantum computing (for now).

---

> ### Author Rebuttal · Authors · 2023-08-09
>
> We appreciate the reviewer taking the time to thoroughly review our paper and provide helpful feedback.
>
> Regarding the use of QRAM, we would like to clarify that prior works (Refs. [50] and [7]) also have exactly the same requirement of QRAM (though under slightly different names). Specifically, Ref. [50] utilizes a "tree-like data structure in QCRAM" as described on Page 7, Line 4. Furthermore, Theorem 9 of Ref. [50] emphasizes the use of "a quantum computer with QCRAM." Ref. [7] also states they employ the same classical variable data structure as Ref. [50] per Lemma 1. In summary, QRAM was commonly assumed in these prior works.
>
> Thank you for catching the typo on Line 285. We will correct this in the revised version.

---

> > ### Comment · Reviewer_8qKg · 2023-08-11
> > **Reply to Authors**
> >
> > Thank you for addressing my concerns in a satisfying way. While I still consider the necessity of the QRAM as a minor weakness, it then seems like a necessary evil that is hard to get rid of in this case (as is the case with many quantum algorithms).

---

> > > ### Author Response · Authors · 2023-08-15
> > > **Thank you!**
> > >
> > > Thanks for your comments!
> > >
> > > We agree that QRAM seems inevitable in designing efficient quantum algorithms for zero-sum games and linear programming solvers.

---

### Official Review · Reviewer_XSuW · 2023-07-05

**Soundness:** 3 good
**Presentation:** 2 fair
**Contribution:** 3 good
**Rating:** 6
**Confidence:** 4

**Summary:**

The paper studies the online learning quantum algorithms for zero-sum games. It proposes the online quantum algorithm for zero-sum games with near-optimal regret, which makes progress to online learning algorithms in the quantum setting. The algorithm quantizes classical algorithms based on the MMW method and incorporates a fast quantum multi-sampling procedure for the Gibbs sampling problem. By developing a sample-based stochastic version of the optimistic multiplicative weight update method, this method achieves a quadratic improvement over classical algorithms and presents a fast quantum linear programming solver as an application.

**Strengths:**

1. The paper addresses a timely and relevant topic in quantum learning theory, proposing the first online quantum algorithm for zero-sum games with near-optimal regret.
2. The algorithm achieves a quadratic improvement over classical algorithms for computing the approximate Nash equilibrium.
3. The paper presents a fast quantum linear programming solver as an application.

**Weaknesses:**

1. The main algorithm is derived from classical methods, and the explanation of the features/contributions of quantum computing in achieving the speedup or advantage could be further elaborated.
2. As a theory paper, it could benefit from a clear and well-motivated example demonstrating the application of the proposed algorithm. Including such an example would help readers better understand the potential practical implications and usefulness of the work.
3. Could the authors discuss whether this online quantum algorithm could be dequantized?
4. In the abstract, the authors claim that "a fast quantum linear programming solver" but haven't identified under which setting.

**Questions:**

Refer to the part about weaknesses.

**Limitations:**

NA.

---

> ### Author Rebuttal · Authors · 2023-08-09
>
> We thank the reviewer for their detailed comments.
>
> For the contributions of quantum computing in achieving the speedup, we would like to explain as follows: to obtain the speedups over prior work, we proposed Sample-based Optimistic Multiplicative Weight Update (SOMWU for short) as the framework for solving zero-sum games. By integrating SOMWU with quantum techniques, we achieve near-optimal regret, surpassing prior quantum approaches at a comparable cost. However, efficiently implementing SOMWU requires a tailored quantum sampling algorithm. Existing methods are insufficient, motivating our novel multi-Gibbs sampler design. We will highlight these ideas in the future version of our paper.
>
> To demonstrate an application of our zero-sum game algorithm, we discuss its use for solving linear programming problems, as noted in the Introduction (Section 1, Page 3). Specifically, we show how our approach can be leveraged as a quantum linear program solver in Section 5 (Page 9), with further details provided in Appendix E (Page 30). We will expand this example in the revised paper to further illustrate the practical utility.
>
> Regarding dequantization, quantum-inspired algorithms usually assume certain conditions of the matrices, such as the low-rank condition (e.g., Ewin Tang. “A quantum-inspired classical algorithm for recommendation systems” and Nai-Hui Chia et al. “Sampling-based sublinear low-rank matrix arithmetic framework for dequantizing quantum machine learning”). It is unclear if our algorithm can be efficiently dequantized, as our problem setting does not assume that the input matrix is sparse or low-rank. Therefore, while dequantization may be possible in certain constrained cases, it is unlikely a classical version could match our performance for general inputs. Further investigation of the dequantization question remains an interesting area for future work.
>
> As for the setting of our quantum linear programming solver, the solver requires quantum oracles to the input matrix and vectors of the linear program, and it outputs a classical value as the approximate optimal value of the program. This setting is standard in quantum algorithms for linear programming. We will expand Section 5 to include the detailed setup of the quantum linear programming solver.

---

> > ### Comment · Reviewer_XSuW · 2023-08-20
> >
> > Thanks for the reply that helps me better understand. I have no further questions.

---

### Official Review · Reviewer_UumW · 2023-07-05

**Soundness:** 3 good
**Presentation:** 2 fair
**Contribution:** 3 good
**Rating:** 6
**Confidence:** 4

**Summary:**

This paper considers the problem of developing quantum, low-regret, algorithms for solving zero-sum games. This paper provides a quantum algorithm which matches the state-of-the-art for solving zero-sum games while improving upon the regret of the associated algorithm. The paper achieves this result by providing a stochastic variant of optimistic mirror descent which they then implement by a quantum algorithm for developing multiple samples from a suitable Gibbs distribution. Additionally, the paper discusses implications and lower bounds for linear programming more broadly.

**Strengths:**

Zero sum games are an important optimization problem in many disciplines. The mere fact that the paper shows how to depart from previous quantum algorithms for solving zero sum games (in using optimistic mirror descent versus mirror descent) while retaining state-of-the-art bounds is interesting and potentially useful for further study of the problem. Additionally, by following this alternative approach the paper obtains improved regret and a type of parallelism in solving the problem. This is an interesting advancement for the study of the problem. Finally, this alternative optimization method gives rise to a different Gibb’s distribution sampling problem than the one considered in prior work. Furthermore, the paper gives an interesting algorithm for solving it which could motivate further study. Finally, the additional discussion of linear programming provided in the paper (though seemingly straightforward) is a nice addition as well.

**Weaknesses:**

It is not clear how substantial the departure from this paper from prior work is, how novel the advancements are, and how complete the comparison of this paper to prior work is. While I think the advancements of the paper are interesting and potentially exciting (as discussed in the strength section), I fear that the paper does not well-articulate them or situate them. For example:

* A comparison of the techniques in the Gibb’s sampling algorithm of this procedure and those in prior quantum-zero sum games algorithm does not seem to be provided. This seems particularly important as the techniques were key to recent advances in quantum algorithms for solving zero-sum-games.
* It is not clear what exactly the benefits of low regret are over prior work and why previous results couldn’t be used to obtain low-regret (e.g. by only considering a subsequence of the iterates taken). I think the low-regret aspect is interesting and this algorithm is potentially more parallel than prior algorithm’s with state-of-the-art complexity and the sampling problems considered may be simpler than in prior work; however I am not sure that the paper argues this clearly.
* It is not clear what obstacles were present to obtaining the result. It would help if the paper clarified whether moving from mirror-descent to optimistic mirror-descent was the main insight and the rest was straightforward (though technical and important) or if further insights were needed.

Some additional comments along this line are given “Questions.”


**Questions:**

My main questions / comments are the items in the previous “Weaknesses” section. Beyond what was written there: How does the Gibb’s sampling compare to prior work? Is it simpler in any way? The same? Is a completely different techniques used? Is there a major insight of the paper beyond the use of stochastic optimistic mirror descent?

Beyond this, below are some more detailed suggestions, questions, and comments:

* Line 1: “for zero-sum games” --> “for solving zero-sum games”
* Line 4: “yielding a quadratic improvement” as written, I am concerned this is suggesting that it is the first such improvement.
* Line 27: “For, this approximation task, online learning becomes significant” – I am not exactly sure what was meant by this.
* Line 33: It might be beneficial to define regret or explain the model more clearly earlier so the improvement over prior work is clearer. (Regret is well-known in general, but when stated for a static optimization problem, the definition is less clear.)
* Line 39: I believe zero-sum games were shown to be solvable in $\tilde{O}(1/\epsilon)$ iterations earlier, e.g. “Prox-method with rate of convergence O (1/t) for variational inequalities with Lipschitz continuous monotone operators and smooth convex-concave saddle point problems” by Nemirovski in 2004. I’m less sure about whether this works in regret context but in any event I believe should be mentioned when discussing the history of solving zero-sum games.
* Line 81: it might be good to clarify whether this is a new feature of this algorithm or not
* Line 130 – 135: it might be good to clarify why prior work could or could not be used for this problem
My main questions / comments are the items in the previous “Weaknesses” section. Beyond what was written there: How does the Gibb’s sampling compare to prior work? Is it simpler in any way? The same? Is a completely different techniques used? Is there a major insight of the paper beyond the use of stochastic optimistic mirror descent?

Beyond this, below are some more detailed suggestions, questions, and comments:

* Line 1: “for zero-sum games” --> “for solving zero-sum games”
* Line 4: “yielding a quadratic improvement” as written, I am concerned this is suggesting that it is the first such improvement.
* Line 27: “For, this approximation task, online learning becomes significant” – I am not exactly sure what was meant by this.
* Line 33: It might be beneficial to define regret or explain the model more clearly earlier so the improvement over prior work is clearer. (Regret is well-known in general, but when stated for a static optimization problem, the definition is less clear.)
* Line 39: I believe zero-sum games were shown to be solvable in $\tilde{O}(1/\epsilon)$ iterations earlier, e.g. “Prox-method with rate of convergence O (1/t) for variational inequalities with Lipschitz continuous monotone operators and smooth convex-concave saddle point problems” by Nemirovski in 2004. I’m less sure about whether this works in regret context but in any event I believe should be mentioned when discussing the history of solving zero-sum games.
* Line 81: it might be good to clarify whether this is a new feature of this algorithm or not
* Line 130 – 135: it might be good to clarify why prior work could or could not be used for this problem


**Limitations:**


This is primarily a theory paper, and it is unclear how applicable this is. However, the weaknesses and questions discussed reflect limitations for which further discussion might be beneficial.

---

> ### Author Rebuttal · Authors · 2023-08-09
>
> We appreciate the reviewer's feedback highlighting opportunities to better articulate the novelty and situate our techniques relative to prior work.
>
> Specifically, we will include a detailed comparison of our Gibbs sampling approach to existing methods from quantum zero-sum game literature.  Our quantum multi-Gibbs sampler is different from the samplers appearing in prior works (Ref. [50] and Ref. [7]).
> The sampler in Ref. [50] is a special case of ours when the sample count $k = 1$; also, the learning algorithm in Ref. [50] only requires the case when $k = 1$.
> In the following we compare the sampler from Ref. [7] with ours:
>
> - Firstly, the two samplers are designed for different goals. We focus on the optimistic multiplicative weight update, while Ref. [7] adopt the non-optimistic version. As a result, we need a good procedure to obtain many samples at a time to reduce the variance. They use the dynamic Gibbs sampler to reduce the amortized time complexity of obtaining a single sample which changes at every step.
> - Secondly, the implementation details are very different. In our quantum multi-Gibbs sampling procedure, we develop a proper version of consistent amplitude estimation to guarantee the correctness of the quantum k-maximum finding. In the dynamic Gibbs sampler, they maintain certain invariants and a hint vector to achieve a good amortized time complexity.
>
> Regarding regret bounds, we will emphasize the concrete advantages of our approach over prior quantum algorithms. Prior works (Ref. [50], [7]) employed the Sample-based Multiplicative Weight Update (SMWU) framework (by Ref. [25]), while we proposed Sample-based Optimistic Multiplicative Weight Update (SOMWU for short) which is a generalization of the Optimistic Multiplicative Weight (OMWU) proposed in Ref. [48]. The regret bounds of these quantum algorithms inherit from their classical framework. However, whether the MWU (or SMWU in Ref. [25]) can achieve $\tilde{O}(1)$ regret in the zero-sum game setting has been open for more than a decade (see, for example, Ref. [19, Future work in Page 3]). To the best of our knowledge, we are not aware whether previous results (Refs. [50], [7]) can be used to directly obtain low regret under the game setting (this is even not known achievable in the classical setting).
>
> The reviewer rightly notes that a core insight is moving from mirror descent to optimistic mirror descent. In addition, the following insights are crucial in designing our algorithm:
>
> - Our first insight is the sample-based generalization of OMWU, which, to the best of our knowledge, is a new framework for solving zero-sum games.
> - Our second insight is to point out that SOMWU, equipped with quantum computing, is able to achieve near-optimal regret, beating prior quantum approaches while retaining the computational cost. To implement SOMWU efficiently, we need a specifically-designed sampling algorithm, while known quantum samplers do not apply here as they are not efficient enough.
> - Our third insight is the quantum multi-Gibbs sampler, which, we believe, can serve as a basic subroutine in future quantum machine learning algorithms.
>
> Regarding the detailed questions for lines, we will address them as follows:
>
> - We will revise the wording on lines 1, 4, 27, and 81 to improve clarity as suggested.
>
> - We will define regret earlier and explain the model more explicitly to make our paper more understandable (Line 33).
>
> - Thanks for pointing out the work of Nemirovski in 2004, which investigates how to find the saddle point in convex-concave minimax optimization problems. This work is highly related to our work, though it has a non-online problem setting. We will cite the paper and discuss the proximal method and other histories of solving minimax optimization problems that are related to solving zero-sum games to properly credit prior work (Line 39).
>
> - We will explain why existing techniques could not achieve our efficiency gains, to motivate the need for new methods (Line 130-135), as noted in the second insight.

---

> > ### Comment · Reviewer_UumW · 2023-08-14
> > **Response to Rebuttal**
> >
> > Thank you for the reply. I believe adding the information in your rebuttal (and better defining / articulating what exactly the low regret setting / problem means for this paper) would all improve it.

---

> > > ### Author Response · Authors · 2023-08-15
> > > **Thank you!**
> > >
> > > Thanks for your further comments.
> > >
> > > We will definitely incorporate our discussion in the rebuttal into future versions of this paper.
> > >
> > > Regarding defining/articulating what exactly the low regret setting/problem means for this paper, we want to explain more as follows:
> > >
> > > - Lower regret means fewer interactions between players when they reach an approximate Nash equilibrium in the game setting and fewer rounds for the algorithm to find a good approximation of the optimal value for the linear programming solver. In previous quantum approaches (Refs. [50, 7]), they all have $\tilde O (\sqrt{T})$ regret; that is, the regret would be bounded by the square root of the number of rounds. By contrast, our logarithmic-regret algorithm has a near-constant bound (up to logarithmic factors) of regret. Thus, our algorithm has a good theoretical dependence on the number of rounds for finding the Nash equilibrium of zero-sum games and for finding an approximate optimal value for linear programming problems.
> > >
> > > We will add these explanations in the revised version of our paper.

---

### Official Review · Reviewer_42Uz · 2023-07-06

**Soundness:** 3 good
**Presentation:** 3 good
**Contribution:** 3 good
**Rating:** 6
**Confidence:** 4

**Summary:**

In this paper the authors consider the task of computing the eps-approximate Nash equilibrium of zero-sum games. In particular, for a matrix A in R^{m x n}, the goal is to find the minimum distributions x,y s.t., max_y x^t A y - min_x x^t A y is at most epsilon, given quantum query access to A.  In this paper the authors are further concerned with the online learning/regret model, i.e., in each round the players need to choose probability distributions (u,v) and their goal is to pick these distributions (u,v) such that it is as close as possible to the optimal choice of eps-Nash equilibrium.

Prior known quantum algorithms for computing the Nash equilibrium ran in time sqrt{m+n}/eps^2.5 but achieved a regret of sqrt{T}  for a T round algorithm. In this paper, the authors show how to obtain a regret of O(1) with the same complexity as sqrt{m+n}/eps^2.5.

**Strengths:**

The main strength of this paper is in a way it solves an important question of computing Nash equilibrium with a O(1)-based regret algorithm.  Using this, they algorithm obtain a state of the art quantum algorithm for linear programming solvers. I think both of these contributions are interesting on their own right!

**Weaknesses:**

As for weakness, i feel there are a couple of weaknesses:

(i) The main techniques used in this paper, aren't very novel. Indeed, they do improve upon prior works and need new tools, but the novelty of the main contributions towards obtaining their speedups doesn't make it a strong contribution. For example, obtaining k copies of a Gibbs state in time sqrt{nk} (compared to the trivial k sqrt{n} algorithm) isn't too surprising (indeed new, but not surprising given Hamoudi's algorithm and also complexity of Grover's search with k marked items).

(ii) The speedup to obtain the regret bound is by quantizing the optimistic  online algorithm. I think there are some subtelties when quantizing this algorithm, but the main ideas involved aren't novel.

Having said the above, i think overall for the ML community, i think the main results and these interesting quantum subroutines above are nice.

**Questions:**

I think the definitions of online learning and regret can be defined much better. In particular, the online learning/regret model for Nash equilibrium is very messily written and I'd have appreciated a better explanation.

---

> ### Author Rebuttal · Authors · 2023-08-09
>
> We appreciate the reviewer identifying potential areas for improvement. However, we believe the core techniques demonstrate non-trivial novelty for the following two points.
>
> First, our framework of Sample-based Optimistic Multiplicative Weight Update is considered for the first time in both classical learning theory and quantum algorithm research. The framework of our Sample-based Optimistic Learning algorithm contains a new and detailed analysis of the regret (see Appendix B, Page 15). The regret analysis differs from the classical counterpart that does not involve sampling; moreover, it also differs from all previous quantum approaches that do not consider the technique of Regret bound by Variance in Utilities (RVU, Definition A.2, Page 15).
>
> Second, while the multi-Gibbs sampler adapts existing ideas of designing quantum samplers like Hamoudi's algorithm (Ref. [29]), creatively extending these methods to efficiently generate independent samples from the Gibbs distribution when only implicitly knowing the parameters represents an innovative application. We propose our novel multi-Gibbs sampler to satisfy the requirement of the sample-based optimistic multiplicative weight update framework, as existing quantum samplers do not. We agree the building blocks of quantum singular value transformation, amplitude estimation, etc. are known. However, we tailored these techniques in novel ways for our application.
>
> In summary, we appreciate the reviewer highlighting opportunities to further demonstrate novelty. In the revision, we will expand the discussion part to better highlight our contribution.
>
> As for the question raised by the reviewer regarding the definitions of online learning and regret for finding Nash equilibria, we agree that concisely explaining these concepts would make our paper more understandable. In short, online (machine) learning studies the situation when data arrives in a sequential order and each time the (meta-)algorithm is required to output a predictor for future data. Regret is a commonly-used performance measure in online learning that compares the losses of predictors produced by the algorithm and the optimal fixed predictor. As a concrete example, the well-known multiplicative weight update algorithm is an online learning algorithm with regret $\tilde{O} (\sqrt{T})$. We will carefully revise Section 2 to provide an easy-to-follow introduction to online learning and regret for finding Nash equilibria.

---

> > ### Comment · Reviewer_42Uz · 2023-08-15
> >
> > I thank the authors for their reply. For the time being, I'll keep my rating and keep in mind your comments for future discussions.

---

> > > ### Author Response · Authors · 2023-08-17
> > > **Thank you!**
> > >
> > > Thank you for reading our response! Please do not hesitate to reach out if you have further questions.

---

### Decision · Program_Chairs · 2023-09-21

**Decision:**

Accept (poster)

**Comment:**

This paper proposes the first online quantum algorithm for zero-sum games with logarithmic regret. In particular, the paper shows how to obtain a regret of $O(1)$ with the same complexity as $sqrt{m+n}/eps^2.5$. We reached a consensus of acceptance for this paper after the discussion. However, the authors should discuss the connections to existing literature (mentioned by all the reviewers, especially reviewer UumW) more carefully for the camera-ready version. I recommend the paper be accepted.